# Empirical upscaling of OzFlux eddy covariance for high-resolution monitoring of terrestrial carbon uptake in Australia.

Chad. A. Burton[1], Luigi. J. Renzullo[1,2], Sami. W. Rifai[3], and Albert. I.J.M. Van Dijk[1]

[1] Fenner School of Environment & Society, Australian National University, Canberra, ACT, Australia.
[2] Bureau of Meterology.
[3] School of Biological Sciences, The University of Adelaide, Adelaide SA, Australia.
*Correspondence to*: Chad Burton (chad.burton@anu.edu.au)

**Abstract.** We develop high resolution (1 km) estimates of Gross Primary Productivity (GPP), Ecosystem Respiration (ER) and Net Ecosystem Exchange (NEE) over the Australian continent for the period January 2003 to June 2022 by empirical upscaling of flux tower measurements. We compare our estimates with nine other products that cover the three broad categories that define current methods for estimating the terrestrial carbon cycle and assess if consiliences between datasets can point to the correct dynamics of Australia's carbon cycle. Our results indicate that regional empirical upscaling greatly improves upon the existing global empirical upscaling efforts, outperforms process-based models, and agrees much better with the dynamics of $CO_2$ flux over Australia as estimated by two regional atmospheric inversions. Our nearly 20-year estimates of terrestrial carbon fluxes revealed Australia is a strong net carbon sink of -0.44 (IQR=0.42) PgC/year on-average, with an inter-annual variability of 0.18 PgC/year and an average seasonal amplitude of 0.85 PgC/yr. Annual mean carbon uptake estimated from other methods ranged considerably, while carbon flux anomalies showed much better agreement between methods. NEE anomalies were predominately driven by cumulative rainfall deficits and surpluses, resulting in larger anomalous responses from GPP over ER. In contrast, we show that the long-term average seasonal cycle is dictated more by the variability in ER than GPP, resulting in peak carbon uptake typically occurring during the cooler, drier Austral autumn, and winter months. This new estimate of Australia's terrestrial carbon cycle provides a benchmark for assessment against Land Surface Model simulations, and a means for monitoring of Australia's terrestrial carbon cycle at an unprecedented high-resolution. We call this new estimate of Australia's terrestrial carbon cycle, "AusEFlux" (Australian Empirical Fluxes).

## 1. Introduction

The global terrestrial biosphere has acted as a net carbon sink, absorbing approximately 29 % of anthropogenic $CO_2$ emissions
each year and thereby mitigating impacts from global warming (Friedlingstein et al., 2022). Australia's vast semi-arid ecosystems play a large and critical role in controlling the inter-annual variability (IAV) of the global terrestrial carbon sink, and are therefore of crucial importance to understand if we are to make reliable predictions about the fate of the global carbon cycle under a warming climate (Ahlström et al., 2015; Chen et al., 2017; Ma et al., 2016; Poulter et al., 2014; Metz et al.,

2023). However, uncertainties in the methods used for quantifying components of the terrestrial biosphere preclude definitive
inferences about the magnitude of Australia's terrestrial carbon sink, the seasonal and inter-annual oscillations, and the drivers
of change in carbon flux variability.

Several methods exist to quantify the spatio-temporal dynamics of the terrestrial carbon cycle. Dynamic Global
Vegetation Models (DGVMs) and Land Surface Models (LSMs) simulate responses of vegetation to changes in climate by
parameterising ecological processes but are limited by several uncertainties that relate to their parametrisations and limited
inclusion of key ecological processes (Kowalczyk et al., 2006; Li et al., 2021; Quillet et al., 2010).  Uncertainties in these
models can lead to large differences in land carbon flux estimates, even where similar models are used (Teckentrup et al.,
2021). For example, over a 17-year period from 2003 to 2019, the Community Atmosphere Biosphere Land Exchange
(CABLE) model extracted from TRENDY v10 estimates Australia's annual mean GPP to be 3.01 PgC/yr (Friedlingstein et
al., 2022) while a regionally forced CABLE run (covering the same period) using a similar model configuration estimates GPP
to be greater than 50 % higher at 4.58 PgC/yr (Villalobos et al., 2022).

Atmospheric inversion methods, which rely upon atmospheric $CO_2$ measurements and an atmospheric transport
model, provide a semi-empirical method for quantifying aspects of the carbon cycle, but their capacity to spatially resolve $CO_2$
fluxes is severely constrained by the sparse observational network of measuring sites (51 sites globally, with only four locations
in Australia) (Rödenbeck et al., 2018). Satellite-based remote sensing of atmospheric $CO_2$ has become possible using the
Greenhouse Gas Observing Satellite (GOSAT) and the Orbiting Carbon Observatory (OCO-2 and OCO-3) satellites (Basu et
al., 2013; Eldering et al., 2017).  This allows for spatially comprehensive monitoring of $CO_2$ sources and sinks over continental
to global scales.  Several global inversion studies have incorporated these datasets, but results over Australia have been
contradictory (Basu et al., 2013; Chevallier et al., 2014; Detmers et al., 2015). Villalobos et al. (2022) conducted a regional
atmospheric inversion over Australia assimilated with OCO-2 data to infer a gridded estimate (~81 km cells) of NEE for 2015-
2019.  They found Australia was a strong annual carbon sink (-0.47 PgC/yr) on average, and that peak carbon uptake occurred
during the cooler, drier months of the austral winter.  Similarly, using an atmospheric inversion of GOSAT satellite
measurements, Metz et al. (2023) found that Australia's seasonal $CO_2$ flux variability coincided with the onset of rainfall after
the dry season, leading to $CO_2$ flux releases during the October-December period, and carbon uptake occurring during the drier
March-September period.  These studies provided valuable insight into the dynamics of Australia's terrestrial carbon cycle,
but their very coarse spatial resolution prevents these approaches from resolving spatially detailed estimates of Australia's
carbon cycle.

A third approach relies on data-driven machine learning (ML) methods to upscale eddy covariance (EC)
micrometeorological tower data from global networks of long-term carbon and water flux measurement sites.  This approach
has the advantage of relying on a denser network of empirical observations than the atmospheric inversion approaches (for
example, the popular FLUXNET2015 dataset contains 206 sites (Pastorello et al., 2020)).  Another advantage of data-driven
ML approaches is their ability to accurately model highly nonlinear relationships to explanatory variables, as is common in
complex environmental systems. Nevertheless, the results of global empirical upscaling products, most notably FLUXCOM

(Jung et al., 2020; Tramontana et al., 2016), are prone to several limitations, including: significantly underestimating the magnitude of the IAV of carbon fluxes, an inability to resolve carbon flux trends (e.g. from $CO_2$ fertilisation), and overestimating the size of the tropical carbon sink (Jung et al., 2020). The global FLUXNET2015 dataset is also biased to the northern hemisphere, which may preclude global upscaling products from making quality predictions in regions that are both underrepresented in the training data, and do not overly conform to northern hemisphere climate dynamics (Baldocchi et al., 2018; Baldocchi, 2020). Over Australia, two FLUXCOM products: 'FLUXCOM-Met' and FLUXCOM-RS', show substantially different mean annual NEE fluxes of -0.23 and -0.05 PgC/yr, respectively (averaged over the period 2003-2015). Furthermore, the annual mean GPP and ER components show a > 60 % difference in magnitude between the two products. IAV of NEE, as estimated by one standard deviation in the fluxes, is also subdued compared with estimates from LSMs and atmospheric inversions.

This lack of agreement between the different approaches to quantifying Australia's land carbon sinks and sources calls into question how well constrained the magnitudes, IAV, temporal trends, and spatial allocations of Australia's land carbon fluxes are. Here we explore the potential for empirical upscaling of the regional "OzFlux" eddy covariance network (Isaac et al., 2017; Beringer et al., 2016; Beringer et al., 2022) to better characterise Australia's terrestrial carbon cycle. Models built on global datasets (and with a strong northern hemisphere bias) will necessarily need to generalise across vastly different climates, ecosystem types, and plant functional traits, limiting their ability to accurately represent ecosystem dynamics in regions where ecosystem responses do not conform to the dominant dynamics in the global dataset. This may especially be the case in Australia where extreme climate variability and evolutionary isolation have created sclerophyllous, evergreen, woody species that do not fit into standard globally predominant plant functional types used by LSMs (Beringer et al., 2016; Beringer et al., 2022; Williams and Woinarski, 1997). Furthermore, Australia's data record of EC flux tower measurements has grown substantially in the intervening years since the inception of the commonly used FLUXNET2015 training dataset. For example, the FLUXCOM product included data from only four EC flux towers over Australia (~43 site-years of data), and the current FLUXNET2015 dataset contains 23 sites equating to ~115 site years of Australian data. Contrast this with the full OzFlux dataset over Australia which, as of January 2022, contains 33 sites and 238 site-years of data. These later years of EC flux tower measurements since 2015 are especially valuable given they have recorded a period of extreme climate variability in Australia such as the historic drought from 2017-2019 (Fang et al., 2021) culminating in the Black Summer bushfires (Byrne et al., 2021), and the subsequent triple La Niña with record breaking rainfall in eastern Australia from 2020-2023. A further advantage of upscaling fluxes at a regional scale is the ability to take advantage of higher-resolution input datasets than is tractable at the global scale, both due to the unavailability and uncertainty of global high-resolution datasets and the computational constraints that attend global upscaling.

Our objectives for this study are as follows:

- Develop an accurate, high-resolution (~1 km) empirical upscaling of Net Ecosystem Exchange (NEE), Ecosystem Respiration (ER), and Gross Primary Productivity (GPP) for Australia covering the period January 2003 to June 2022.

- Evaluate our empirical upscaling of Australian flux data in comparison with LSM, inversion-derived, and global empirical upscaling estimates of the carbon cycle with the aim of identifying consiliences between datasets that may point to the correct dynamics of Australia's terrestrial carbon cycle.
- Assess if the upscaling approach can offer new insights into Australia's carbon cycle, and/or affirm if the upscaling can replicate known biogeochemical controls on the carbon cycle.

## 2. Data & Methods

### 2.1 Data

#### 2.1.1 CO$_2$ flux tower data

We used monthly fluxes of NEE, GPP, and ER produced by the OzFlux (https://ozflux.org.au/) regional network of eddy covariance flux towers. These data are processed to Level 6 and are freely accessible through the Terrestrial Ecosystem Research Network THREDDS portal (https://dap.tern.org.au/thredds/catalog/ecosystem_process/ozflux/catalog.html (TERN, 2023). All site data used in this study was version "2022_v2", and in instances where both "site-pi" and "default" versions of the datasets were available, we utilised the "default" datasets. Twenty-nine of the 33 freely available sites were selected. The four sites that were excluded showed strong landscape heterogeneity within the flux tower footprint, insufficient temporal duration, or non-representative landcover (e.g., almond farms). A summary of the selected sites and their locations is shown in Figure A1. The Level 6 OzFlux data used in this study provides two separate estimates of constituent carbon fluxes derived from two methods for partitioning NEE into its component fluxes of GPP and ER. This study uses the 'SOLO' data version which is calculated using a data-driven nocturnal respiration approach for partitioning where respiration is modelled using an artificial neural network driven by air temperature, soil temperature, and soil water content. (a full description of the SOLO partitioning method is provided within (Isaac et al., 2017)). We trained ML models with the flux data at a monthly temporal resolution using 2,825 monthly observations, equating to 235 site-years.

#### 2.1.2 Gridded explanatory variables

The variables in Table 1 were selected for inclusion in the modelling framework as they were considered to cover most of the expected climate and landscape controls on the terrestrial carbon cycle in Australia. MODIS derived datasets were temporally resampled to monthly resolution using the mean of all clear observations within a given month and reprojected onto a 1-km x 1-km geographic grid for prediction using averaging resampling techniques. The static variables of landcover fractions and vegetation height were also resampled to 1 km resolution using the average of all pixels within a 1 km grid. The 1 km grid was selected to match the coarsest native resolution explanatory variables, namely the climate datasets. The training procedure uses data extracted from the same 1-km gridded data (using the pixel located over the EC tower).

**Table 1: Gridded feature layers used in the modelling framework to train and predict terrestrial carbon fluxes over Australia.**

| Explanatory Variable (abbreviation) | Description | Data Source & Reference |
|---|---|---|
| Land Surface Temp. (LST), Normalised Difference Water Index (NDWI), Kernel Normalised Vegetation Index (kNDVI), | A suite of MODIS derived products characterising the land surface responses to climate. In addition, fractional anomalies are calculated for the kNDVI variable to account for disturbances from fire or land-use change. Fractional anomalies are calculated against a long-term climatological mean from 2003-2021. | MODIS Collections MCD43A4 & MOD11A1 (version 6.1) downloaded from Google Earth Engine: https://developers.google.com/earth-engine/datasets/catalog/modis |
| Average Air Temp. (Tavg), Vapour Pressure Deficit (VPD), Incoming Shortwave Radiation (srad), Total Precipitation (rain) | ~1 km resolution gridded climate products based on topographically conditional spatial interpolation of Australia's extensive network of weather stations. In addition, fractional anomalies are also calculated for all variables except VPD. In addition to monthly fractional rainfall anomalies, three-, six-, and twelve-month cumulative fractional rainfall anomalies are added to help characterise memory and lag in the carbon response to water deficit. | ANUClimate: https://dapds00.nci.org.au/thredds/catalogs/gh70/catalog.html (Hutchison et al., 2014) |
| LST minus Tavg (LST-Tair) | The subtraction of air temperature from land surface temperature is indicative of vegetation canopy moisture stress | Derived from MODIS LST and ANUClimate Tavg |
| Fraction Trees (trees), Fraction C4 grass (C4_grass), Fraction Grass (grass), Fraction Bare (bare), | Per-pixel fractions of trees, grass, and bare derived from temporal decompositions of MODIS NDVI into persistent and recurrent fractions. An estimate of the proportion of C4 grass is also included. These variables are static and represent conditions in 2020. | Correspondence (Donohue, 2021) |
| Vegetation Height (VegH) | A per-pixel estimate of vegetation height in metres. This variable is static and represents the average vegetation height from 2007-2010. | Accessible from https://dapds00.nci.org.au/thredds/catalog/ub8/au/LandCover/OzWALD_LC/catalog.html (Liao et al., 2020) |

### 2.1.3    Comparison datasets

Datasets included for comparative purposes cover the three current categories of methods for estimating the exchange of terrestrial carbon with the atmosphere: process-based models, empirical upscaling of eddy covariance data, and atmospheric inversions. Observation-based GPP products derived from light-use-efficiency methods and solar-induced fluorescence are also included for completeness. Where possible, datasets are processed and plotted in their native resolutions to avoid introducing errors from spatially resampling finer-resolution datasets to very coarse resolutions (or vice-versa). The exceptions to this are the higher-resolution MODIS-GPP and DIFFUSE-GPP products (described below) which were resampled to 1 km resolutions to match the resolutions of our ML upscaling product. A summary table of all the comparison datasets is available in the appendix (Table A1).

#### 2.1.3.1    Process-model simulations

We compared our results with two runs of the CABLE model. The first was a regional, fine resolution (0.25º) offline run forced by Australian regional climate drivers that follows the protocol from Haverd et al. (2018) but with land use remaining static at the year 2000 (hereafter referred to as CABLE-BIOS3). CABLE-BIOS3 net biosphere production (NBP) includes GPP and autotrophic and heterotrophic respiration, but does not include fire disturbances, harvest, erosion or export of carbon through rivers (a fuller description of the set-up is outlined in Villalobos et al. (2022)). A second CABLE run was extracted from the TRENDY v10 ensemble (Friedlingstein et al., 2022), hereafter referred to as CABLE-POP. This dataset has a spatial resolution of 1º, is forced by global climate data and NBP includes additional fluxes from fire emissions and land use change.

#### 2.1.3.2    FLUXCOM

Our regional ML upscaling product is compared with the well-known global ML upscaling product, FLUXCOM (Jung et al., 2020; Tramontana et al., 2016). FLUXCOM is built using similar machine learning methods to those used in this study, though trained on the global FLUXNET2015 dataset. Two products are available, FLUXCOM-RS was trained exclusively on MODIS remote sensing data, and FLUXCOM-RS+METEO (FLUXCOM-Met hereafter) which is trained on climate reanalysis data and climatological remote sensing data (Jung et al., 2020). For FLUXCOM-Met, we use the multi-model mean of the ERA5-based product. Both RS-METEO and RS products are assessed here and were downloaded at monthly temporal resolution from the Max Planck Institute for Biogeochemistry (https://www.bgc-jena.mpg.de/geodb/projects/Home.php, last access 13/01/2023).

#### 2.1.3.3    Atmospheric Inversions

A regional inverse modelling product, produced by Villalobos et al. (2022) was included for comparison as it provides a wholly independent measure of NEE. This regional inversion estimates carbon fluxes over the Australian continent from 2015-2019 by assimilating of carbon-dioxide measurements from the Orbiting Carbon Observatory-2 (OCO-2) satellite. The product is

provided at ~81 km spatial resolution and monthly temporal resolution (available for download from https://zenodo.org/record/6649768). NEE in this dataset includes fire emissions and fossil fuel emissions, so to facilitate better comparisons fossil fuel emissions were subtracted from the NEE time-series. A second regional satellite-assimilated atmospheric inversion from Metz et al. (2023) is also included. This timeseries represents the spatially averaged net flux of $CO_2$ over the Australian TRANSCOM region (which includes New Zealand). Therefore, the time-series is only shown where
Australian-wide spatially averaged time-series are plotted, and some differences between time-series may be attributable to the inclusion of the New Zealand land mass in the estimate.

### 2.1.3.4 Observation-based GPP products

We compare our GPP estimates with a suite of observation-based GPP products: the MODIS Terra GPP product (MOD17A2H), based on a per-biome light-use efficiency approach (Running et al., 2015); the GOSIF GPP product, generated
through a data-driven approach based on OCO-2 SIF soundings, MODIS remote sensing data, and meteorological reanalysis data (Li and Xiao, 2019); and DIFFUSE GPP which is based on total and diffuse irradiance and the fraction of shortwave irradiance absorbed by foliage (Donohue et al., 2014). All datasets are averaged to monthly temporal resolution, and MODIS-GPP and DIFFUSE-GPP are spatial resampled to 1-km grid cells by averaging the pixels within each 1 km pixel grid.

### 2.1.4 Fire emissions

Fire emissions were added to our estimates of NEE from the Global Fire Assimilation System version 12 (GFASv12) (Kaiser et al., 2012). Daily fire emissions are temporally resampled to monthly totals by summing daily values.

### 2.1.5 Bioclimatic regions

Bioclimatic regions used for separating fluxes into specific ecosystems were identical to those defined in Haverd et al. (2013) and include six bioclimatic classes: tropics, savanna, warm temperate, cool temperate, Mediterranean, and desert (Fig. 9a).

## 2.2 Methods

### 2.2.1 Empirical ML upscaling

The most common ML models implemented in the literature on empirical upscaling of EC data are random forest regression, support vector regression, model tree ensembles, piecewise regression models, and artificial neural networks (Verrelst et al., 2015). Random forest (RF) regression has proven itself to be the go-to model for many remote sensing-based studies owing to
its high accuracy, robustness to over-fitting, scalability, and easy to configure hyperparameters (Belgiu and Drăguț, 2016). In recent years, gradient-boosting decision tree (GBDT) learning algorithms have also proven to be highly accurate and robust to overfitting (Chen and Guestrin, 2016; Wei et al., 2019). Here, rather than rely on any one ML method, we rely on both RF and GBDT methods to develop an ensemble of predictions.

Beyond the ML algorithm used, there are numerous other sources of uncertainty associated with the empirical upscaling of EC flux tower data. Epistemic uncertainties arise from the limitations of the training data (e.g., biases in the locations sampled), and uncertainties in the features used for training, as well as the hyperparameters used during model optimisation. In addition to these reducible (or at least quantifiable) epistemic uncertainties, aleatoric uncertainties arise from the uncertainties of the eddy covariance measurements themselves (Isaac et al., 2017), along with the non-deterministic dependencies between variables (Hüllermeier and Waegeman, 2021). Here we attempt to account for a portion of the empirical uncertainty by iterating the training data and the models used for fitting. During model fitting, two randomly selected EC sites are removed from the training data and both a GBDT model (from the python package LightGBM (Ke et al., 2017)) and a RF model are fit on the remaining data (hyperparameter optimization is conducted on every fit using a random grid search technique with 250 iterations). The reason we selected two sites to remove per iteration was because we felt it balanced the need to significantly alter the training dataset per iteration, while not overly degrading the quality of the model by removing too much data. This procedure is repeated 15 times to increase the likelihood of every site being removed from the training dataset, resulting in 30 unique models. These 30 models are used to generate 30 gridded estimates for each of the variables modelled (GPP, ER, and NEE). In the results that follow, we report the interquartile range of these 30 predictions as our envelope of uncertainty, and the 'best-estimate' as the median of the ensemble predictions.

The overall modelling framework is summarised in Figure 1. Each flux is independently modelled, and therefore there is no inherent exact mass balance between GPP-ER and NEE. The same predictor variables were used for modelling each flux, so the resulting products originate from a consistent set of drivers. All processing and modelling steps described in the method sections have been thoroughly documented within a series of Jupyter Notebooks, available with the assets of this paper.

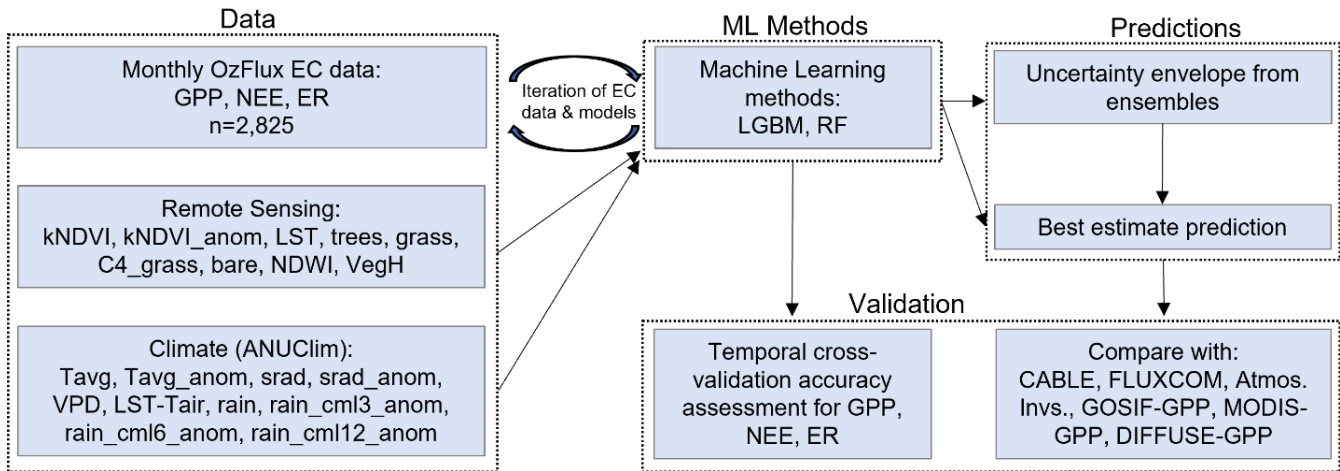

**Figure 1: A flow chart showing the modelling framework for creating gridded estimates of GPP, ER, and NEE for the Australian continent.**

### 2.2.2 Model evaluation

The accuracy of each ML model in the ensemble was assessed using a nested, time-series-split cross-validation approach (Fig. 2).  This approach ensured minimal data leakage between training and testing sets, while still allowing the algorithm to 'see' all the sites during training; a desirable feature in the cross-validation technique due to the relatively limited number of sites (n=29), with some ecosystems sampled by only one or two flux towers (e.g., alpine regions, cereal cropping).  Five outer cross-validation splits are performed, with each split containing 20 % test data from every site (as a discrete length of time equal to 20% of the total length of the dataset; i.e if a site contained 10 years of data, then testing was conducted on five iterations of two-year continuous periods), while the remaining 80 % of the data is used for training.  Five 'inner' cross-validation splits were conducted to optimise the hyperparameter selection for the outer loop. Using a nested approach to cross-validation prevents using the same data to tune model parameters as the model is tested on, and thus prevents creating overly optimistic cross-validation scores (Cawley and Talbot, 2010). Across the five outer cross-validation splits, all samples in the dataset were tested.  Mean Absolute Error (MAE) and the coefficient of determination ($R^2$) are reported to assess the accuracy of the fit for each of the variables modelled.  The cross-validation scores reported in the results section summarise the train-test splits of all 30 model fits.  Throughout the remainder we use the terms 'observed' and 'predicted' to refer to in-situ measurements from EC towers and the predictions, respectively. We also use the convention of negative NEE values referring to net carbon uptake by the land surface.

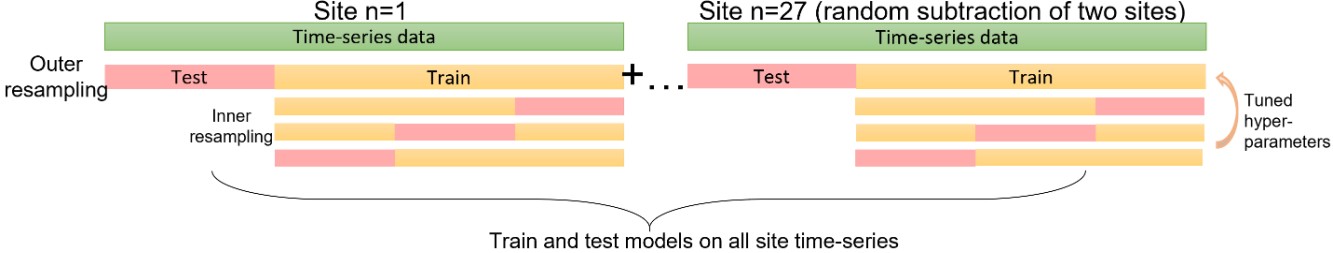

**Figure 2: A schematic representation of a single cross-validation split using a nested, time-series cross-validation procedure.  Five outer splits and five inner splits were conducted per model iteration. For each split, models are trained on data from every site included in that model iteration (i.e., 80 % of every site) and tested on a continuous period for every site (i.e., 20 % of each site). For each subsequent split, the test period is moved forward in time.**

In addition to evaluating the overall predictive capacity using temporal cross-validation, we also perform an intercomparison between the results of this study and similar products covering Australia. This is performed through scatter plots of modelled vs observed fluxes for several products (statistics for comparison are MAE and $r^2$ – the square of Pearson's correlation), through comparison of the mean seasonal cycles disaggregated by bioclimatic region, and through the assessment of annual anomalies.  It is important to note that NEE calculated through empirical upscaling of EC flux tower data is conceptually distinct from inversion-based NEE and process-model NBP.  The addition of fire-emissions to our estimates of

NEE narrows the conceptual distance between the estimates, and where a conceptual difference still applies, we contend that fluxes from other sources are unlikely to be large enough to warrant the additional complexity of their inclusion.

## 3. Results

### 3.1 Cross-validation performance

Temporal cross-validation results revealed a comparatively high degree of agreement between observations and predictions (Fig. 3). As for other regional and global upscaling products, GPP and ER were predicted with better skill than NEE. GPP scored a $R^2 = 0.91$ and MAE = 19.4 gC/m$^2$/month. For ER, $R^2 = 0.89$ and MAE = 15.8 gC/m$^2$/month, while for NEE, $R^2 = 0.68$ and MAE = 17.9 gC/m$^2$/month. To understand how well the predictions reproduce annual mean fluxes, and the per-biome predictability of fluxes, we produced scatter plots comparing the annual mean fluxes of the EC flux tower sites with the annual

mean fluxes of the median of the prediction ensemble (Fig. 3d-f). Regardless of biome, annual mean fluxes were exceptionally well reproduced by the median of the ensemble with the 'all-data' fit closely matching the one-to-one line. The climatological seasonal cycles of NEE at each of the EC sites were also very well reproduced (Fig. A2).

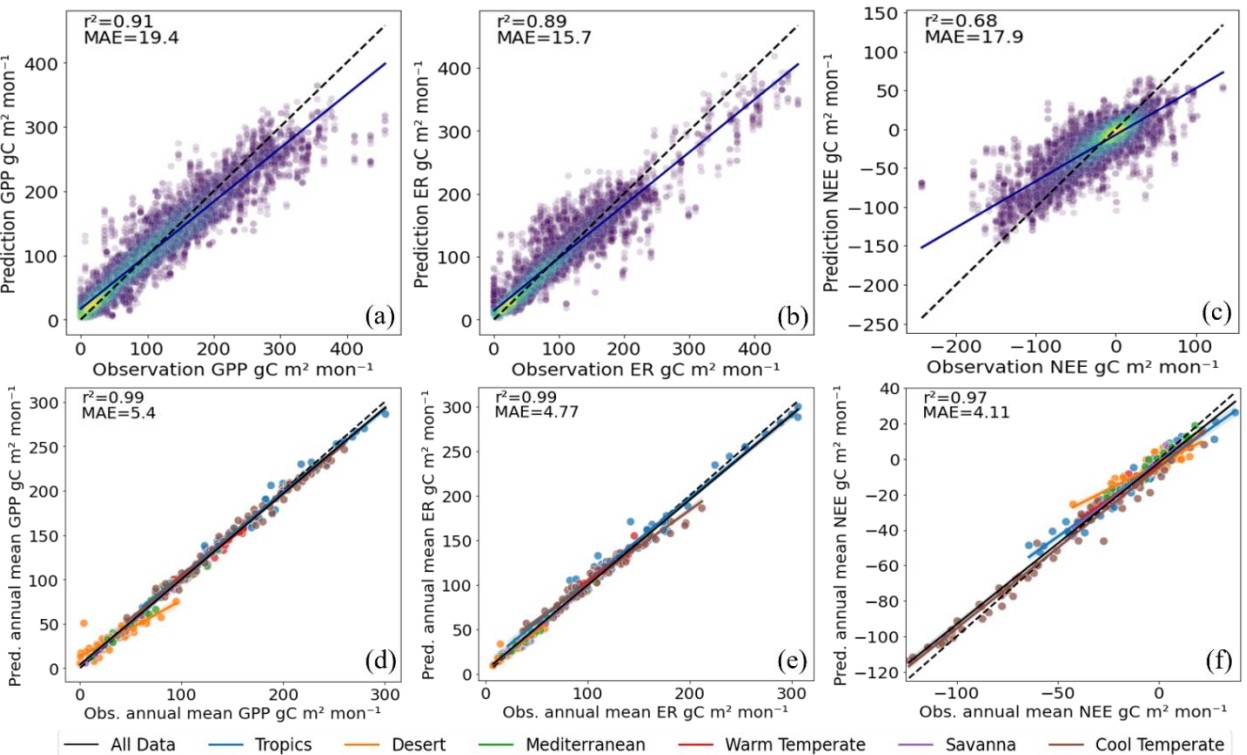

**Figure 3. Pooled temporal cross-validation results for EC flux tower sites: scatter plots of observed and predicted monthly (a) GPP, (b) ER and (c) NEE, with heat colours indicating data density; and scatter plots of observed and predicted annual mean (d) GPP, (e) ER, and (f) NEE, with colour coding indicating bioclimatic regions, as shown in Figure 9a.**

Scatter plots showing the trend and strength of the relationships between EC flux tower observations and modelled values for other products can be found in the Appendices (Fig. A3). The EC flux tower values are compared with the nearest pixel in each product, and the products have been reprojected to match the resolution of CABLE-BIOS3 (~25 km). Only those products with a reasonably high spatial resolution have been compared with the flux tower (i.e., CABLE-POP, FLUXCOM-Met, and the OCO-2 inversion have been excluded), and comparisons were only made for periods where all included products have data. Most products perform reasonably well at predicting GPP (Fig. A3a-f). Typically, products show an overestimation of small GPP and ER values, and an underestimation of large values, except for CABLE-BIOS3's which overestimates GPP and ER across the distribution. CABLE-BIOS3's estimates of NEE showed almost no correlation with EC flux tower observations, recording a $r^2$ of 0.04 (Fig. A3j). The FLUXCOM NEE products performed considerably worse than the cross-validation scores reported in this study (Fig. A3k-l).

### 3.2 Feature importance

To understand which explanatory variables most impacted flux predictions, feature importance plots were produced using the Shapley Additive Explanations (SHAP) Python library (Lundberg & Lee, 2017). Shapley values represents the average marginal contribution of a feature value across all possible coalitions (Lundberg et al., 2020). The feature importance bar plots of Figure 4 show the top five ranked features for each modelled flux, ranked in descending order with the most important variables at the top. These plots were derived by calculating the mean absolute SHAP values for each feature in each model iteration, and subsequently averaging those values across all the models in the ensemble. Flux predictions were strongly influenced by the remote sensing variables of kNDVI and NDWI which respond to canopy density, health, and water status. Solar radiation and average air temperature were the most important climate variable across the fluxes. The land cover variables of vegetation height and fraction of trees also proved important for flux predictions.

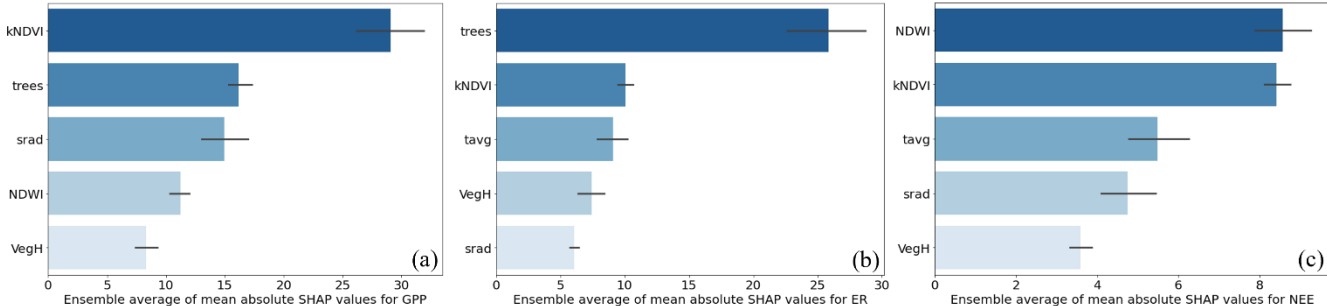

**Figure 4. Shapley additive explanation (SHAP) feature importance plots. (a) GPP, (b) ER, (c) NEE. The plots summarise feature importance across all models in the ensemble by first calculating mean absolute SHAP values for each feature in each model, and then averaging those values across all the models in the ensemble. The error bars show the 95 % confidence interval.**

SHAP dependence plots for kNDVI along with the four principal climate drivers in the model (temperature, rainfall, solar radiation, and VPD) aid in the interpretation of feature importance (Fig. 5; these plots were created using a single optimised GDBT model fit on all the training data). In these plots, the feature values are plotted against their corresponding SHAP values, and the dots are coloured by, in the case of the climate drivers, kNDVI, and in the case of kNDVI, by the values of the feature that has the strongest interaction effect with kNDVI. A strong interaction between two variables produces a

distinct vertical colour gradient. The dependency plots for the climate features are coloured by kNDVI as it aids in approximately disaggregating the influence of climate on carbon fluxes between the wetter, cooler, and high kNDVI coastal fringe regions of the Australian continent from the drier, warmer, lower kNDVI regions of Australia's (semi) arid interior. In the dependence plot for kNDVI (Fig. 5a), solar radiation shows a clear interaction effect. Where kNDVI is low (< ~0.2), increasing solar radiation produces predictions of GPP that are relatively lower than in regions with higher kNDVI. Solar

radiation was the third most important feature in the prediction of GPP (Fig. 4a), and high kNDVI regions had a greater light sensitivity than low kNDVI regions (Fig. 5b).

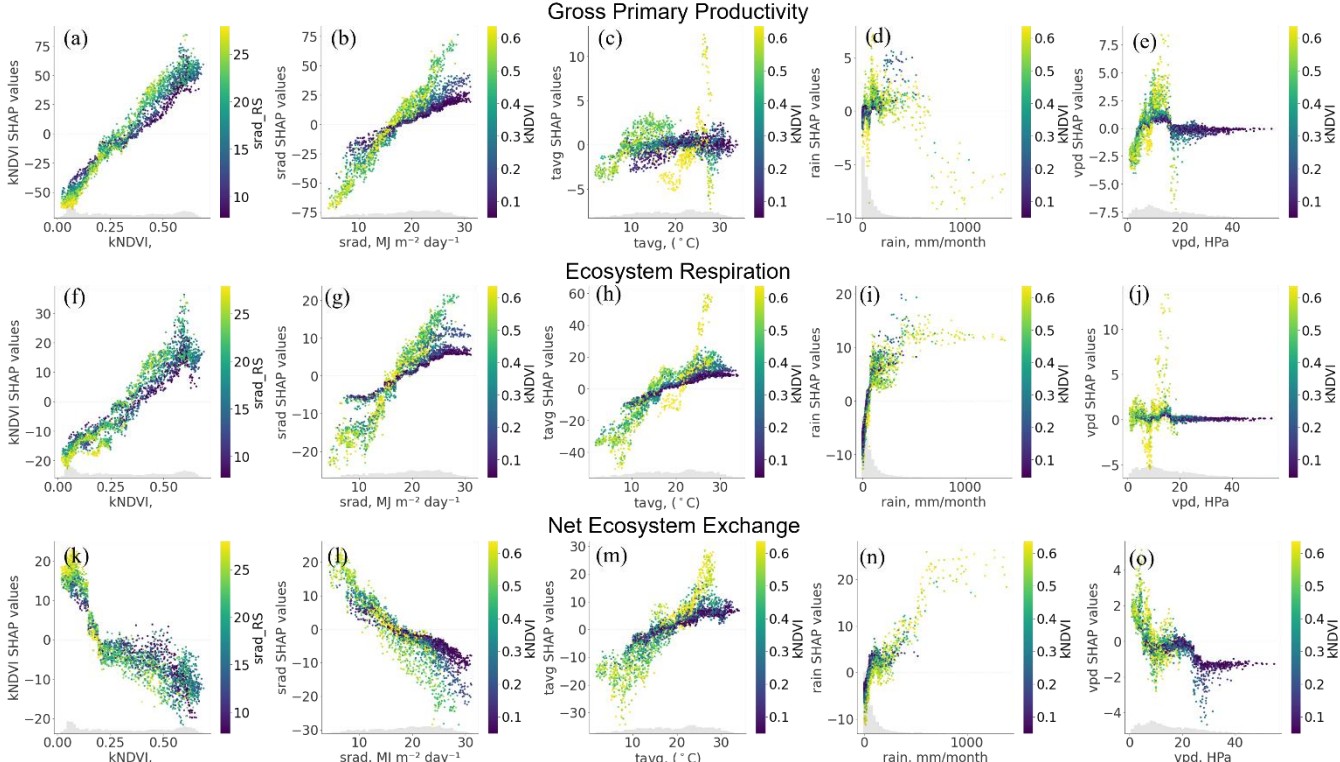

**Figure 5. SHAP dependency scatter plots for kNDVI, along with the four principal climate features (solar radiation, air temperature,**
**rainfall and VPD). In the case of (a,f,k) the SHAP values are coloured by the feature with the largest interaction effect, while the climate variable SHAP values are coloured by their interaction with kNDVI. Note that the y-axis scale is different for each sub plot.**

Solar radiation and kNDVI were also key predictors for ER, following similar relationships as GPP, but the overall amplitude of increase is less (Fig. 5f & 5g). ER also sees a greater influence from air temperature (Fig. 5h) and rainfall (Fig. 5i) than GPP, where higher values of these variables increased predicted rates of ER. In the case of air temperature, in areas of high kNDVI the rate of ER increase was greater than in low kNDVI regions. Rates of ER respiration increase sharply with increased rainfall, but for low kNDVI, predictions of ER increase at a more rapid rate than for high kNDVI (Fig. 5i).

Relationships between features and NEE predictions are more difficult to interpret given the likelihood of complex interaction effects when modelling the carbon balance (NEE) *versus* modelling only ER or GPP. The most important features for the NEE predictions are kNDVI and NDWI, average air temperature, and solar radiation (Fig. 4c). Increasing solar radiation typically resulted in more negative NEE predictions (greater uptake of carbon) (Fig. 5l). The rate of increase in carbon uptake under increasing solar radiation is lower where kNDVI is low, while regions of high kNDVI see a much greater sensitivity to increases in solar radiation. Increasing air temperature tends to result in more positive NEE predictions (Fig. 5m), though the relationship does not follow a simple trajectory. For high kNDVI, temperature increases at the highest end of the distribution (>~25ºC) result in a strong positive rate of change in NEE predictions (i.e., greater release of carbon). For very low kNDVI, temperature changes have a much more modest impact on NEE.

### 3.3 Prediction uncertainties

The coefficient of variation between the thirty ensemble members provides a spatial indication of uncertainty in $CO_2$ flux predictions (Fig. 6). We use a non-standard definition of the coefficient of variation where the median absolute deviation between the long-term annual means of each ensemble member were divided by the median of the ensemble annual means, expressed as an absolute value. Both GPP and ER show comparatively low variability across predictions, where the greatest coefficient of variation values is found in the arid interior (Fig. 6a and 6b). NEE shows stronger variation between ensemble members in some of the arid regions of the north-west, the savannah regions of western Queensland, and the agricultural regions of the Western Australian wheat belt and the Murray-Darling Basin (MDB) (Fig. 6c and 6d). In the case of the arid and savanna regions, the uncertainty coincides with areas where annual mean NEE is close to zero, so small deviations in predictions can result in high relative uncertainty (refer to the annual mean flux map in Figure 8g). However, in parts of the aforementioned agricultural regions, uncertainty is both high in relative and absolute terms (again refer to Figure 8g).

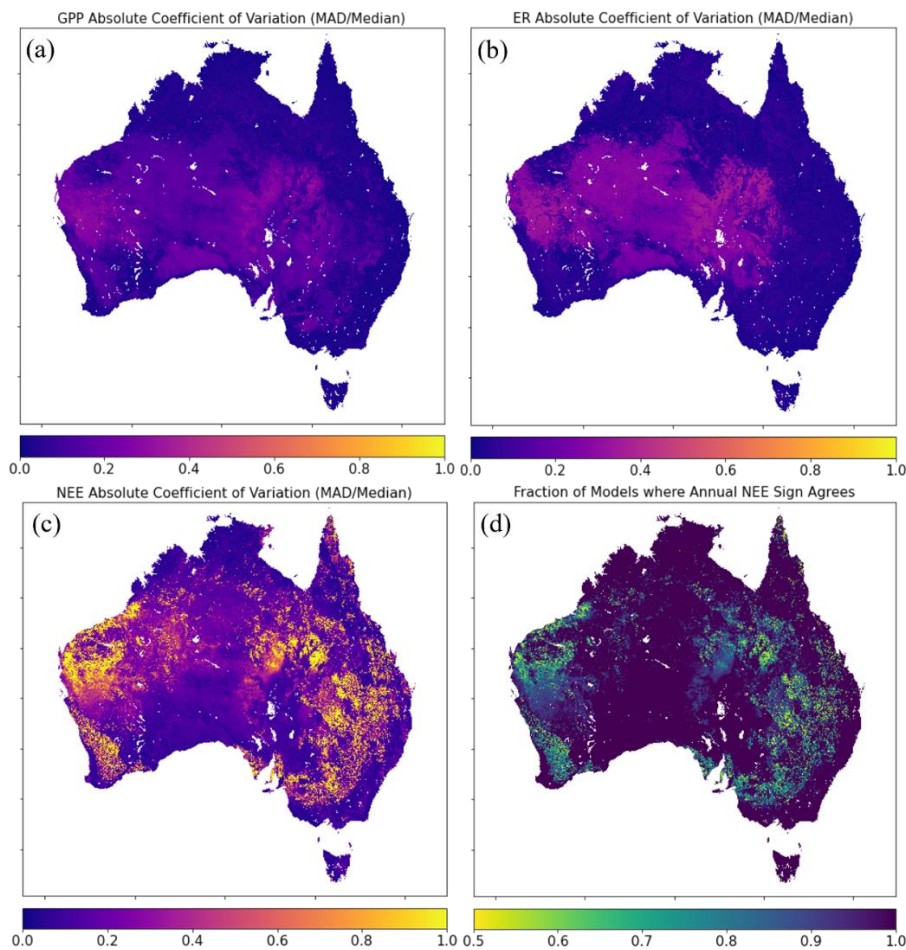


**Figure 6. Prediction uncertainty estimated from iterating EC flux tower data and model type: (a-c) displays the absolute coefficient of variation for (a) GPP, (b) ER, and (c) NEE, defined as the median absolute deviation between all ensemble members, and the median of the ensembles, expressed as an absolute value. (d) shows the fraction of ensemble members where the sign of annual mean NEE (positive or negative) agrees, i.e., if all ensemble members agree on the sign of NEE then the values is one, and if positive and**

**negative estimates are each produced by half of the members, then value is 0.5**

### 3.4  Upscaling results and comparison with other products

#### 3.4.1    Annual mean and IAV of carbon fluxes across Australia

We adopted the model ensemble median as our best estimate, and the interquartile range (IQR) of estimates as a measure of uncertainty. During 2003 to 2022 Australia's terrestrial ecosystems were a strong net carbon sink on an annual mean basis


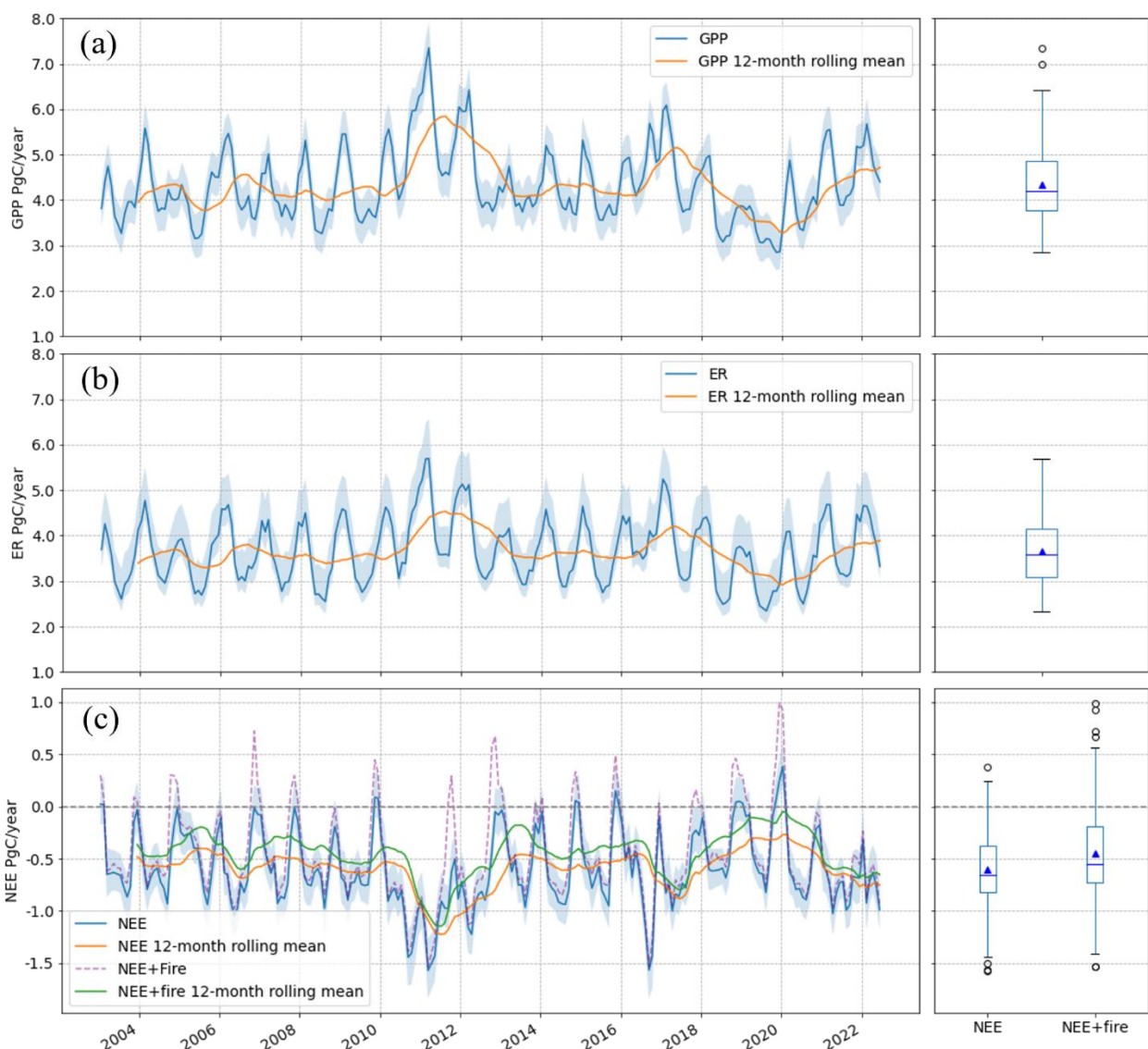

**Figure 7. Monthly carbon fluxes summed across Australia from 2003 to 2021. (a) GPP, (b) ER, (c) NEE). Shading around time-series shows the interquartile range of the prediction ensembles, and the solid blue line shows the median of the ensemble predictions. Orange lines shows the 12-month running mean of the median model. Box plots are based on the median model prediction and show the long-term mean (green triangle), median (line within box) and interquartile ranges (boxes) averaged over the entire time series. c) also shows NEE after adding fire emissions (green line), as estimated by the GFASv12 product.**

of -0.44 (IQR=0.42) PgC/year (Fig. 7c) (including fire emissions). IAV defined as one standard deviation of the annual mean timeseries is 0.18 PgC/year and the average seasonal range of NEE is 0.85 PgC/year. The annual mean estimates of NEE from this study show a greater terrestrial carbon uptake than any of the LSMs or FLUXCOM products, while the regional atmospheric inversion (which also includes fire emissions) predicts a very similar annual mean carbon uptake of -0.47 PgC/year (though this is assessed over a much shorter period than the other products). IAV of NEE for the other products

ranges from 0.06 PgC/yr for FLUXCOM-Met, to 0.26 PgC/year for the OCO-2 inversion. The GOSAT-inversion conducted by Metz et al. (2023) estimated IAV of 0.207 PgC/yr across the Australia TRANSCOM region. CABLE-BIOS3 also shows comparatively high IAV of 0.23 PgC/year (Fig. 9f). The per-pixel plots of Figure 8g-i show how annual NEE fluxes are

spatially allocated. The strongest carbon sinks are seen along the forested coastal regions of the eastern seaboard from western Tasmania to northern New South Wales, the south-west corner of Western Australia including the southern part of the Great Western Woodlands, and the tropical part of the Northern Territory. The regions of strongest IAV in NEE are in the savanna regions of northern Australia, the intensive agricultural regions of the MDB, and the Channel Country of south-west Queensland and into South Australia where episodic river basins such as the Coopers and Meullers Creek periodically fill

during anomalously large rainfall events (Fig. 8h). The climatological 'month-of-maximum' NEE plot in Figure 8i shows the month during which NEE typically achieves its most negative value (greatest carbon uptake), and the plot shows clear delineations along bioclimatic regions.

Annual mean GPP across Australia averaged 4.25 (0.91) PgC/year, with an IAV of 0.50 PgC/year and an average seasonal range of 1.47 PgC/yr (Fig. 7a). Averaged over Australia, our estimate of GPP closely approximates that of GOSIF

and MODIS, with the uncertainty envelope encompassing these two products. In contrast, DIFFUSE, FLUXCOM, and CABLE-POP report lower estimates (Fig. 9a). The IAV between products varies substantially with both FLUXCOM products showing the lowest IAV in GPP (FLUXCOM-Met: 0.13 PgC/year, FLUXCOM-RS: 0.23 PgC/year), while this study and CABLE-BIOS3 (0.78 PgC/year) display the strongest IAV.

ER averaged 3.64 (1.01) PgC/year (Fig. 7b), with an IAV of 0.34 PgC/year and an average seasonal range of 1.56

PgC/year, notably higher than GPP. Agreement between products is generally poor, though the long-term mean of FLUXCOM-Met and this study agree (Fig. 9b). CABLE-BIOS3 show the most IAV in ER (0.56 PgC/year), while the two FLUXCOM products record very low IAV, with FLUXCOM-RS equal to 0.07 PgC/year, and FLUXCOM-Met 0.09 PgC/year.

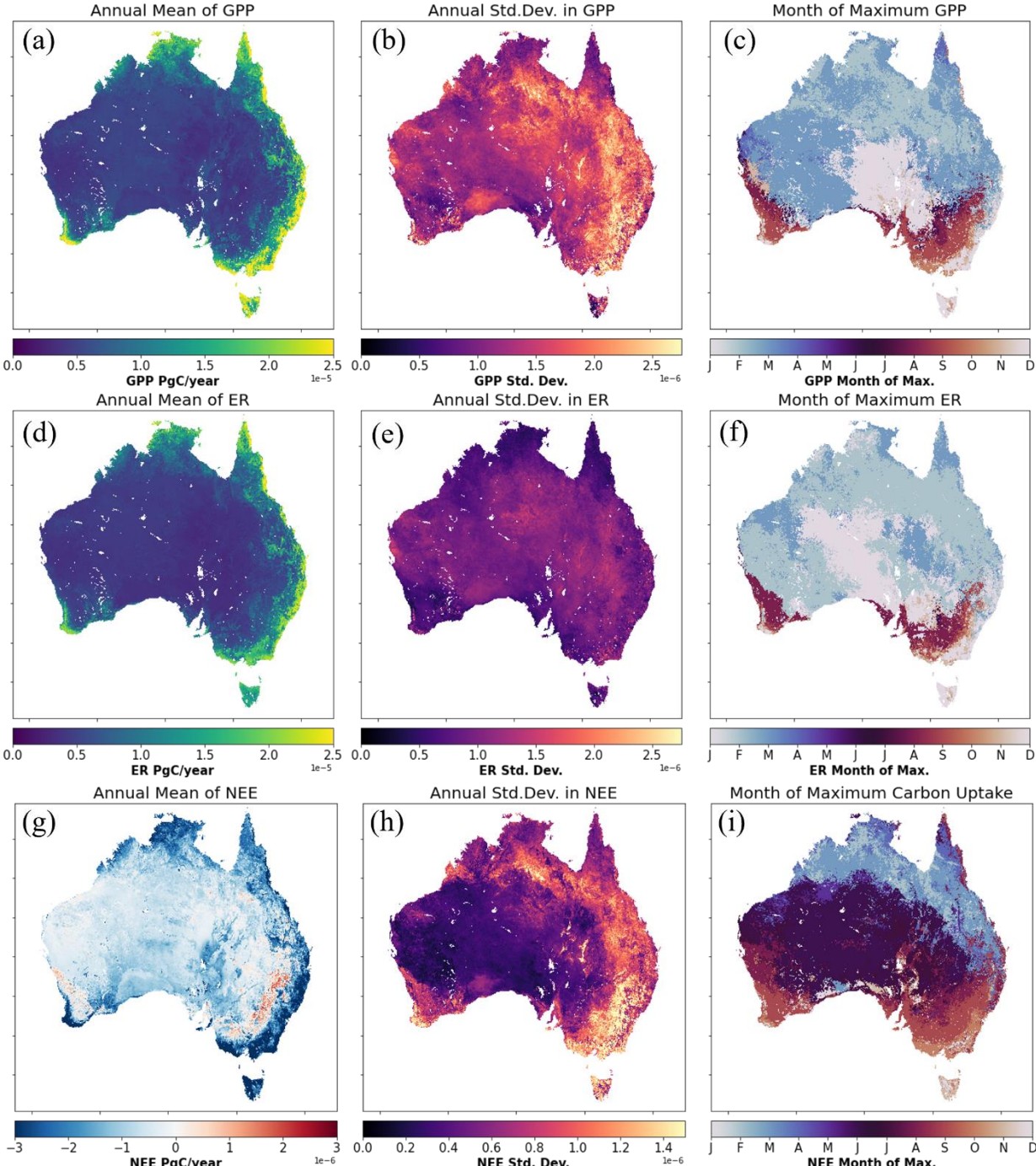

**Figure 8. Per pixel summaries derived from the median of the prediction ensemble. Annual means fluxes of GPP (a), ER (d), and NEE (g), Standard deviation in annual mean fluxes of GPP (b), ER (e), and NEE. Climatological month of maximum flux, GPP (c), ER (f), and NEE (i).**


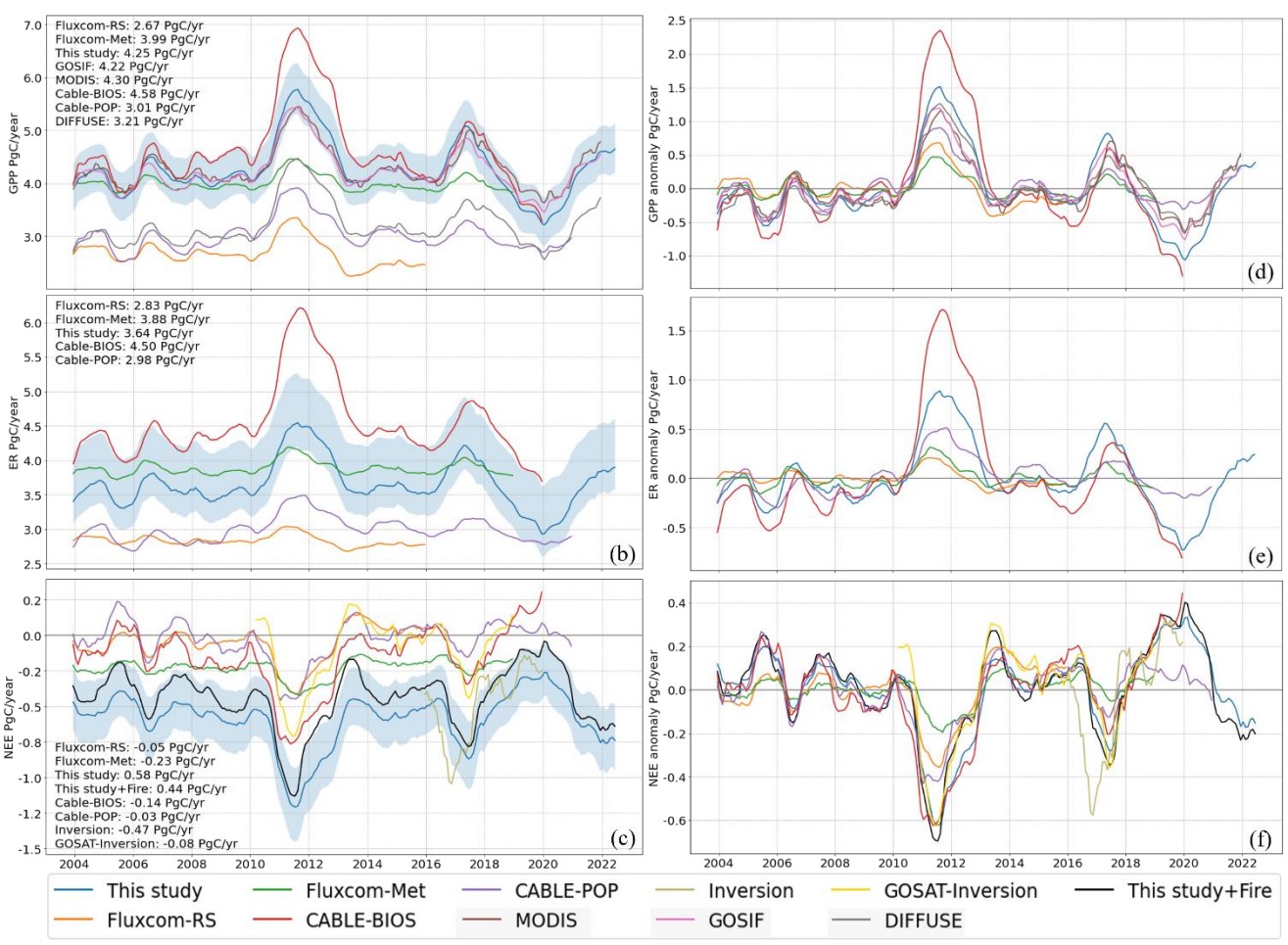

**Figure 9. Twelve-month rolling mean terrestrial carbon fluxes from a suite a products covering Australia, compared with this study. Right-side plots (d-f) show the anomalies of the left-side plots (a-c), where the monthly anomalies are calculated using a climatology that starts in 2003 and ends at the maximum length of the available time-series for each product. The numbers in the left-side plots show the long-term annual mean flux for a given product. Blue shading around 'This Study' shows the interquartile range from prediction ensembles.**

### 3.4.2    Climatological carbon fluxes

Figure 10e-g shows the climatological seasonal cycles of the component terrestrial fluxes summed across Australia (climatologies were calculated starting in 2003 and extending over the full remaining length of the time-series for each product). The seasonal cycle of this study's NEE differs substantially from those of the LSMs and FLUXCOM-Met (Fig. 10g). According to our results, a climatological peak in terrestrial carbon uptake occurs for Australia during the cooler, drier months of March-September. Examination of the equivalent plots for GPP (Fig. 10e) and ER (Fig. 10f), shows that concomitant increases in ER during periods of peak GPP mean that the time of greatest primary production across Australia (December - March) are not coincident with peak carbon uptake. This result contrasts with the findings of the LSMs and FLUXCOM-Met which show peak carbon coinciding with peak GPP in Austral summer (Fig. 10g). Despite displaying a

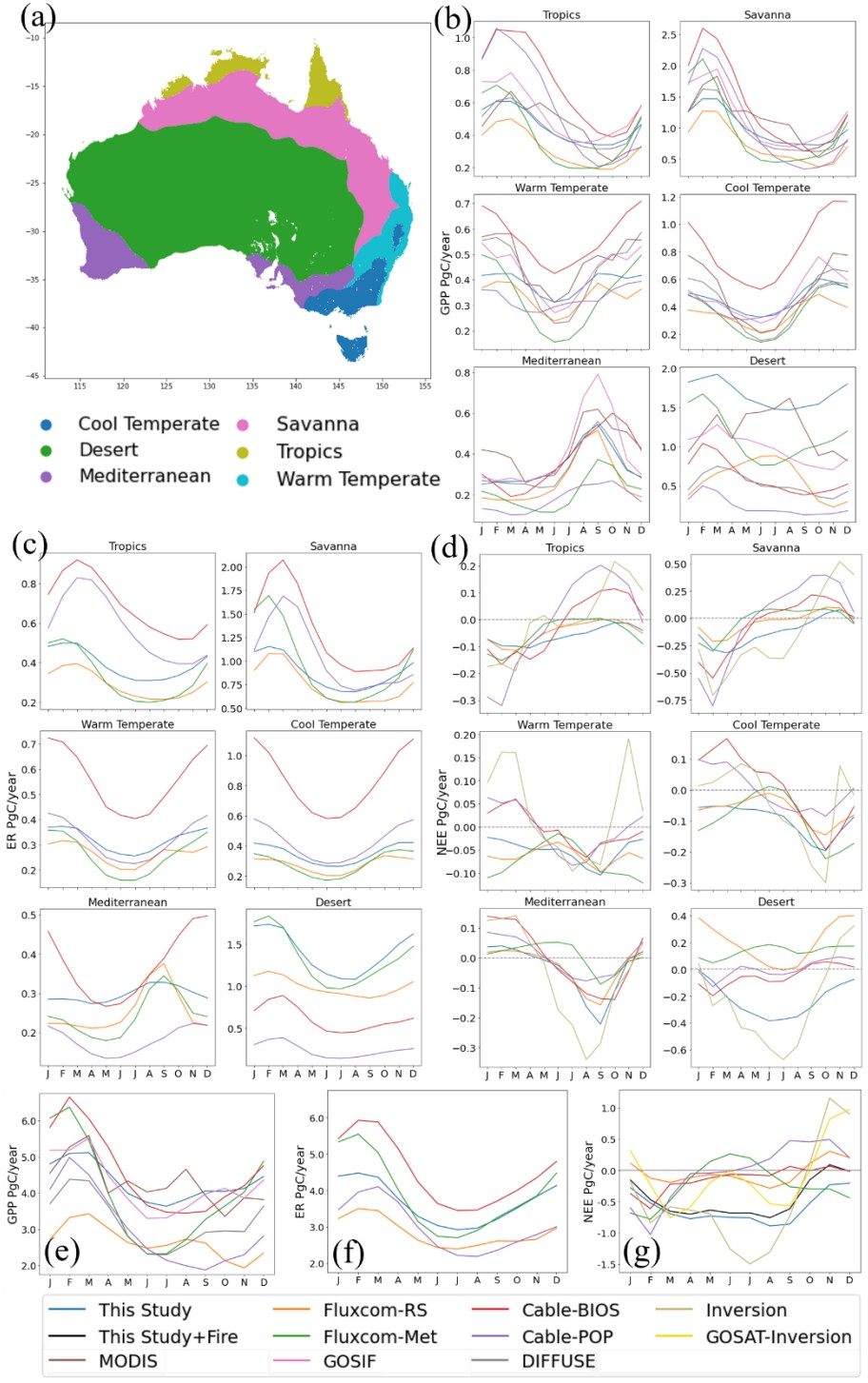

**Figure 10. Climatological seasonal cycles. (a) Map of bioclimatic regions. (b-d) Bioregion specific annual climatological seasonal cycles for GPP, ER, and NEE, respectively. (e-f) Annual climatological seasonal cycles averaged across Australia.**

greater amplitude of seasonal variability, the NEE seasonal cycle of the regional OCO-2 inversion largely matches our estimate. The GOSAT-Inversion also displays similarities with this study and the OCO-2 inversion. However, the GOSAT-inversion shows a second peak in July – it is unclear from the dataset provided if this might be due to the inclusion of New Zealand in the analysis area.

Breaking the fluxes down into bioclimatic zones (Fig. 10a-d), we can observe two processes that predominately dictate the typical seasonal pattern of NEE in Australia. Firstly, seasonal variations in ER in the desert region (peak-to-peak amplitude = 0.66 PgC/yr) exceed GPP variations (amplitude = 0.46 PgC/yr). Beginning in March and extending through the autumn and winter period, ER declines more rapidly than GPP resulting in enhanced carbon uptake during this period. Secondly, in the savanna region we observe a sharp response in ER following the end-of-dry-season rainfall events that exceed the response from GPP, resulting in a net carbon pulse to the atmosphere in the Oct-Dec period (fluxes from these regions are re-plotted in Figure A5a to enhance interpretability). The interaction between these two processes likely explains most of the seasonal variation in Australia's terrestrial carbon cycle, and is responsible for peak carbon uptake in Australia occurring in the autumn-winter months, while the carbon sink tends to be weakest during the Oct-Dec period.

We found that the largest discrepancies between products also occurs in the desert region (Fig. 10a). The LSMs, FLUXCOM-Met, GOSIF, and this study all report GPP peaking in February-March, with the nadir of GPP occurring during the May-Sept period (Fig. 9b). On the other hand, MODIS-GPP and FLUXCOM-RS show an inverted climatology to the other products that are unlikely to be accurate given the monsoonal climate drivers in the region with >70 % of the typical annual median rainfall falling between November and April (Bowman et al., 2010). The CABLE-POP model appeared to significantly underestimate both GPP and ER in desert regions (Fig. 10b-c). This may explain why the Australia-wide seasonal NEE curve from CABLE-POP (Fig. 10g) does not align with the results of this study despite a similar spatial pattern in the month-of-maximum NEE flux plot (Fig. A4). The desert and savanna regions typically contribute the most to annual fluxes in other products, but CABLE-POP's NEE fluxes are comparatively more influenced by the savanna and tropical regions. This is most likely due to CABLE-POP's representation of vegetation cover fractions over inland Australia which show the desert region as entirely bare (Teckentrup et al., 2021). FLUXCOM-RS follows a similar trajectory in the Australia-wide NEE to that of our estimate, though with considerably less seasonal amplitude (Fig. 10g). Examining the bioclimatic zones, we see that this is mainly due to an incorrect GPP seasonal cycle in the desert region, combined with a very low amplitude in the seasonal cycle of ER in the desert (Fig. 10c). The seasonal cycle of FLUXCOM-Met is markedly different from FLUXCOM-RS. The per-biome fluxes from FLUXCOM-Met appear more realistic than those FLUXCOM-RS, but produce an inverted Australia-wide NEE seasonal cycle to our estimate (Fig. 10g). This is due to greater amplitude declines in seasonal GPP compared with ER, especially in the warm and cool temperate regions.

## 3.5 Drivers of carbon flux anomalies

As a simple means for interpreting the drivers of carbon flux anomalies, temporal Pearson correlations between carbon flux anomalies and climate anomalies (respective to 2003-2021 averages) for each bioclimatic zone were conducted (Table 2).

Correlations were calculated per-pixel and then averaged over the bioclimatic zone. Caution in interpreting the results is warranted as the terrestrial carbon cycle is intrinsically complex, and nonlinear. With that caveat: for GPP, ER, and NEE,
cumulative rainfall anomalies almost universally correlate most strongly with carbon flux anomalies. In the case of NEE, across all bioclimatic regions monthly rainfall anomalies were insignificantly correlated. Yet, the cumulative rainfall anomalies proved to be the strongest correlate (where a cumulative rainfall surplus resulted in negative NEE anomalies i.e., greater carbon uptake). In the case of the desert region, correlations of monthly rainfall anomalies jumped from a statistically insignificant r-value of -0.08 to a strong significant correlation of -0.50 for six-month cumulative rainfall anomalies (Table 2), similar scores
were found for the savanna region. Correlations for non-lagged monthly rainfall anomalies in the savanna and desert regions were both much higher for ER than for GPP, suggesting ER responds more quickly to wetting than GPP in the arid and semi-arid regions of Australia.

**Table 2. Temporal Pearson correlations between carbon flux anomalies, climate anomalies, and kNDVI anomalies. Every flux and**
**climate variable anomaly are based on a 2003-2021 baseline. The highest correlation for each flux and bioclimatic zone is shown in bold (for the climate variables only, kNDVI correlations are ignored)**

**BIOCLIMATIC REGION**

| FLUX | Variable | Tropics | Savanna | Warm Temperate | Cool Temperate | Mediterranean | Desert |
|------|----------|---------|---------|----------------|----------------|---------------|--------|
| **GPP** | Rainfall | 0.17 | 0.27 | 0.21 | 0.15 | 0.25 | 0.39 |
| | Rainfall Cml-3 | 0.28 | 0.46 | 0.51 | 0.41 | 0.48 | 0.66 |
| | Rainfall Cml-6 | **0.33** | 0.54 | **0.57** | **0.47** | **0.57** | **0.78** |
| | Rainfall Cml-12 | 0.26 | **0.59** | 0.50 | 0.44 | 0.52 | 0.74 |
| | Air Temp. | -0.01 | -0.36 | -0.25 | -0.11 | -0.23 | -0.36 |
| | Solar Rad. | -0.23 | -0.43 | -0.28 | -0.16 | -0.29 | -0.45 |
| | kNDVI | 0.86 | 0.88 | 0.88 | 0.81 | 0.84 | 0.80 |
| **ER** | Rainfall | 0.45 | 0.49 | 0.54 | 0.45 | 0.60 | 0.55 |
| | Rainfall Cml-3 | 0.39 | 0.54 | 0.67 | 0.58 | 0.69 | 0.68 |
| | Rainfall Cml-6 | 0.38 | 0.62 | **0.67** | **0.59** | **0.72** | 0.78 |
| | Rainfall Cml-12 | 0.22 | **0.63** | 0.60 | 0.56 | 0.68 | **0.79** |
| | Air Temp. | 0.07 | -0.31 | -0.17 | 0.06 | -0.14 | -0.28 |
| | Solar Rad. | **-0.52** | -0.59 | -0.56 | -0.38 | -0.55 | -0.54 |
| | kNDVI | 0.68 | 0.78 | 0.81 | 0.69 | 0.71 | 0.75 |
| **NEE** | Rainfall | 0.12 | -0.02 | 0.09 | 0.05 | 0.07 | -0.08 |
| | Rainfall Cml-3 | -0.13 | -0.31 | -0.30 | -0.23 | -0.22 | -0.42 |
| | Rainfall Cml-6 | -0.25 | -0.40 | **-0.41** | **-0.33** | **-0.32** | **-0.50** |

| | | | | | | |
|---|---|---|---|---|---|---|
| Rainfall Cml-12 | **-0.34** | **-0.49** | -0.36 | -0.29 | -0.30 | -0.41 |
| Air Temp. | 0.15 | 0.35 | 0.31 | 0.25 | 0.28 | 0.44 |
| Solar Rad. | -0.09 | 0.15 | 0.00 | 0.00 | 0.01 | 0.20 |
| kNDVI | -0.69 | -0.79 | -0.70 | -0.67 | -0.68 | -0.57 |

## 4. Discussion

Through our iterative modelling framework, we identified the largest uncertainties in the flux estimates as occurring in the
semi-arid to arid interior, and in the cropping regions of Western Australia and the Murray-Darling Basin (Fig. 6). A limitation
of the OzFlux network is the necessarily limited repeat spatial sampling of all main land cover types. Furthermore, not each
bioclimatic region is equally well represented, leading to biases in the sampling. For example, desert and xeric ecosystems
cover nearly half of the Australian land mass, but less than 10% of the sites are located in these regions (Beringer et al., 2016).
Australia's expansive cropping ecosystems are also under-represented. The limited representation of these systems in the
training data is likely why we found comparatively high uncertainty in these regions (Fig. 6). Further uncertainty in the
cropping regions may also be due to the heterogeneity of crop types and agricultural practices that may not be represented in
our feature layers, and the potentially large carbon exports as agricultural commodities. Given the Australian Government's
emphasis on emission offsetting through changes in agricultural practices and human-induced regeneration of native woody
vegetation, especially in drier regions (Dceew, 2023), new EC sites in cropping regions and in the (semi) arid rangelands areas
of New South Wales, Queensland, and Western Australia might help reduce uncertainties in AusEFlux and expand the
evidential basis for carbon sequestration through (re-)vegetation (Macintosh et al., 2022). Given the changing climate
conditions of Australia, it is vital to maintain the current OzFlux infrastructure so that future changes to climate-carbon
interactions are monitored at the continental level through iterative retraining of the AusEFlux model as new data is collected.

Owing to the limitations introduced by the spatial sampling of the OzFlux network, it is very challenging to effectively
cross-validate terrestrial carbon fluxes in a manner that we could confidently claim accurately estimates the true map accuracy.
This is why we also rely heavily on an intercomparison between products, as we believe the convergence of results from
multiple, independent lines of evidence tells us more about the true nature of Australia's terrestrial carbon cycle than any given
cross-validation method. We are encouraged by the convergence of our results with the GPP estimates from MODIS, GOSIF,
and CABLE-BIOS3 as each of these products applies a different method to quantifying GPP. ER is harder to effectively
validate through a convergence of studies as only FLUXCOM (similar methods to ours) and CABLE provide estimates of ER.
However, the scatter plots of Figure A3 demonstrate that CABLE tends to overestimate ER fluxes, while FLUXCOM-RS
tends to underestimate ER fluxes. AusEFlux estimates of ER lie between these two estimates (Fig. 9b), perhaps indicating that
our estimate of ER is an improvement on the other methods. NEE offers the prospect of independent validation as the satellite
assimilated atmospheric inversions are a wholly independent measurement of NEE (though they still contain significant

uncertainties owing to the uncertainties in the satellite $CO_2$ measurements themselves, along with the atmospheric transport model used). Which is why we include the two most recent regional-scale inversions in our intercomparisons. Though mean NEE varied between our estimate and those of the GOSAT atmospheric inversion, anomalies and the seasonal cycle show better agreement than with other methods. We take this to be evidence that our regional empirical upscaling of the OzFlux network provides a better estimate of Australia's net terrestrial carbon cycle than the global empirical upscaling product, FLUXCOM, which to-date has been the only product available of its type for Australia. Our study showed that increasing the diversity of flux tower sites beyond the small Australian set used in global products improved the quality of carbon flux estimates. We cannot predict whether the same might hold for other underrepresented regions, which mostly coincide with the global south, or whether the isolated evolution of Australia's ecosystems also plays a role.

We found evidence that Australia is, on average, a stronger annual carbon sink than previous CABLE LSM and FLUXCOM estimates have concluded. Our estimate of the long-term annual mean carbon sink over Australia (-0.44 PgC/yr) is higher than those reported by any study besides the regional OCO-2 inversion (-0.47 PgC/yr). We take the consilience between our estimate and the OCO-2 inversion's; the fact that 25 out of the 29 OzFlux EC sites used here report strong annual mean carbon sinks (Figure A7), and the theoretical argument that ML predictions tend to produce good estimates of the mean as evidence that Australia's status as a comparatively strong net carbon sink is robust. Carbon flux anomalies show better agreement between diverse methods, with our estimate, CABLE-BIOS3, and the GOSAT Inversion all largely agreeing on the timing and magnitude of NEE anomalies. The largest annual anomaly, the 2010-11 La Niña anomaly of -0.70 PgC/year reported here (based on a 12-month rolling mean) also aligns well with the -0.77 PgC reported by Ma et al. (2016) and the -0.79 PgC anomaly reported by Poulter et al. (2014). The OCO-2 Inversion, our study, CABLE-BIOS3, and the GOSAT-Inversion also converge on a NEE IAV of ~0.2 PgC/yr (the range among these products is 0.18 to 0.26 PgC/year). Cross-validation showed that our predictions generally underestimate large NEE fluxes (both positive and negative fluxes, Fig. 3). Thus, it is fair to assume that the inter-annual (and seasonal) variability in NEE should be larger than the estimate reported by this study, and perhaps the larger variability of the inversions is closer to the truth. This study is consistent with other studies in showing NEE anomalies in Australia are driven by a greater response of GPP over ER to anomalous rainfall periods (Ahlström et al., 2015; Ma et al., 2016; Poulter et al., 2014; Haverd et al., 2016; Trudinger et al., 2016; Teckentrup et al., 2021; Fig. 9). This is especially the case where rainfall anomalies are cumulative. The strong correlations between cumulative rainfall anomalies and NEE anomalies provides some additional support to the study of Cranko Page et al. (2022), who showed that the inclusion of rainfall lags increased the predictability of site-level NEE in Australia. Australia contributes substantially to the IAV of the global terrestrial carbon sink, an important advantage of our high-resolution dataset is that it allows us to identify and monitor fine-resolution hotspots of IAV (maps showing greater detail are shown in Figure A8).

We have shown that climatological peak terrestrial carbon uptake in Australia occurs in the Austral autumn and winter months owing mostly to more rapid declines in rates of ER compared with GPP over the arid regions of Australia. Concomitant increases in ER during times of high GPP mean that periods of peak primary production do not necessarily coincide with peak carbon uptake on a seasonal basis. This finding agrees with Renchon et al. (2018) at the Cumberland Plains EC flux tower

site, where the forest was a $CO_2$ sink in winter and a source in summer due to larger seasonal amplitudes in ER. Similarly, Metz et al. (2023) found that seasonal rainfall in semi-arid regions after the dry season drives pulses of heterotrophic respiration that precede the GPP response, leading to net carbon uptake not beginning until March. Cleverly et al. (2013), in a site-based study of a semi-arid acacia woodland in central Australia, observed that the first large springtime storms following the dry season resulted in rapid pulses of ecosystem respiration owing to an uptick in moisture limited microbial decomposition of photodegraded litter and flushing of $CO_2$ from soil pore spaces through infiltration. Our results confirm that ER over the savanna region responds quickly to seasonal rainfall events at the end of the dry-season, while GPP responds more slowly resulting in carbon pulses to the atmosphere during the Oct-Dec period. Correspondingly, we also find non-lagged correlations between monthly rainfall climatologies and climatological ER stronger than those for GPP over the semi-arid regions of Australia (Fig. A6). Seasonal fires in the Savanna region contribute to this carbon pulse as more intense, late dry-season (Aug-Oct) fires lead to an earlier net carbon pulse to the atmosphere and larger peak emissions (Fig. A5b).

An advantage of this approach over other methods is its computational efficiency, and, owing to the mature architecture of the OzFlux infrastructure, the ability to programmatically ingest updated or new EC datasets to further refine models. Thus, there is an opportunity for AusEFlux to be incorporated into an annually produced national estimate of Australia's terrestrial carbon fluxes. Any annually produced 'bottom-up' estimate of Australia's terrestrial carbon fluxes could also serve as a compliment to the Global Carbon Project's aims of annually reporting the carbon balance of the world (Papale, 2020). Through regular updating of this dataset, the ecosystems that play an outsized role in controlling Australia's mean carbon sink and contribute substantially to its IAV can begin to be systematically monitored for change.

While our estimate provides a step-forward in our means for assessing the complex, seasonal, and interannual dynamics of Australia's carbon cycle, future work can improve upon this current effort. Firstly, we aim to extend AusEFlux further back in time through the inclusion of satellite observations from the AVHRR and Landsat missions. However, this effort will inform a separate study as it will require addressing cross-sensor calibration issues. A longer record of empirically derived terrestrial carbon fluxes will assist in defining robust environmental baselines from which future changes to the carbon cycle can be assessed. Secondly, new or improved feature layers can be incorporated as they become available (e.g., time-varying estimates of the percentages of trees, grass and bare). And lastly, we aim to explore the prospects of ecological forecasting (Dietze et al., 2018) of the terrestrial carbon cycle as forecasts may be possible where forecasts of the climate are sufficiently detailed.

## 5. Conclusions

We show that regional empirical upscaling can improve considerably upon existing global upscaling products, outperform existing LSMs, perform similar to or better than other empirical GPP products, and replicate the dynamics of $CO_2$ flux over Australia as estimated by two regional atmospheric inversions. Our estimate suggests Australia was a strong carbon sink (2003-2022 average) with an annual mean uptake of -0.44 (0.42) PgC/year, has an IAV of 0.18 PgC/year, and an average

seasonal amplitude of 0.85 PgC/yr. Estimates of the annual mean carbon uptake from other methods varied considerably, and only our study and the OCO-2 inversion agreed. However, carbon flux anomalies showed much better agreement between methods. NEE anomalies were predominately driven by cumulative rainfall deficits and surpluses, resulting in larger anomalous responses from GPP over ER. In contrast, the long-term average seasonal cycle is dictated more by the variability in ER than GPP, resulting in peak carbon uptake typically occurring during the cooler, drier Austral autumn and winter months. Our new estimates of Australia's terrestrial carbon cycle fluxes improve upon our understanding of the magnitudes, seasonal cycles, and processes governing Australia's terrestrial carbon cycle and provides a new benchmark for assessment against future LSM developments, and a means for high-resolution monitoring of Australia's terrestrial carbon cycle.

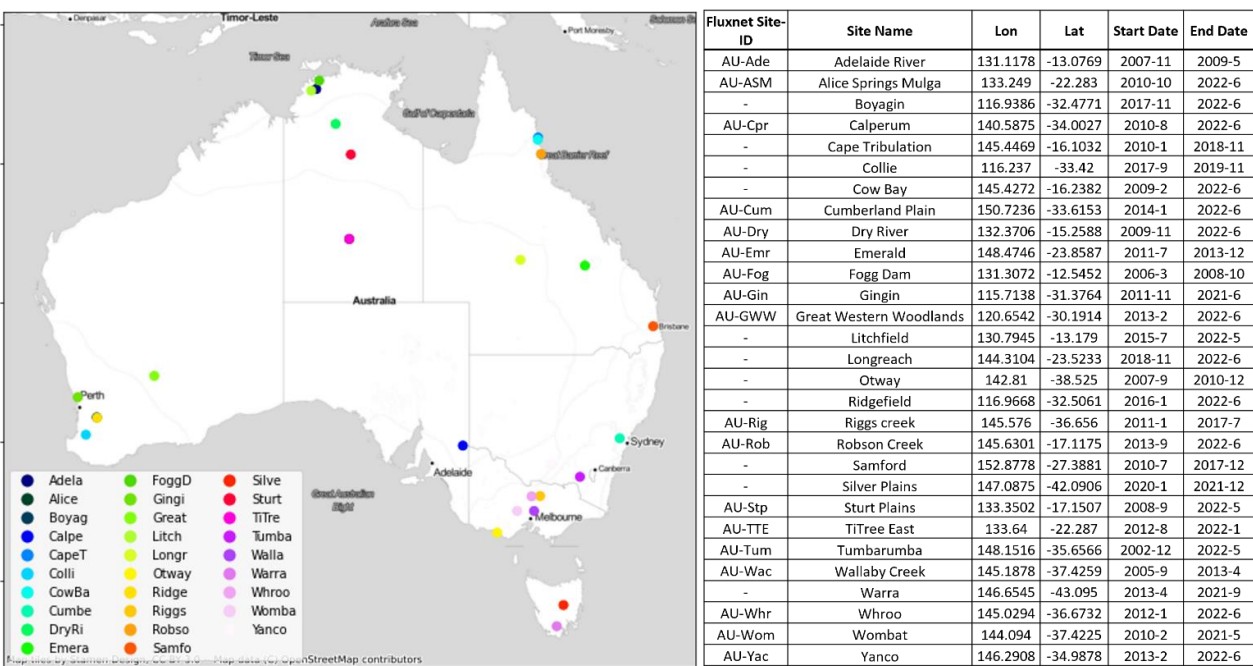

| Fluxnet Site-ID | Site Name | Lon | Lat | Start Date | End Date |
|---|---|---|---|---|---|
| AU-Ade | Adelaide River | 131.1178 | -13.0769 | 2007-11 | 2009-5 |
| AU-ASM | Alice Springs Mulga | 133.249 | -22.283 | 2010-10 | 2022-6 |
| - | Boyagin | 116.9386 | -32.4771 | 2017-11 | 2022-6 |
| AU-Cpr | Calperum | 140.5875 | -34.0027 | 2010-8 | 2022-6 |
| - | Cape Tribulation | 145.4469 | -16.1032 | 2010-1 | 2018-11 |
| - | Collie | 116.237 | -33.42 | 2017-9 | 2019-11 |
| - | Cow Bay | 145.4272 | -16.2382 | 2009-2 | 2022-6 |
| AU-Cum | Cumberland Plain | 150.7236 | -33.6153 | 2014-1 | 2022-6 |
| AU-Dry | Dry River | 132.3706 | -15.2588 | 2009-11 | 2022-6 |
| AU-Emr | Emerald | 148.4746 | -23.8587 | 2011-7 | 2013-12 |
| AU-Fog | Fogg Dam | 131.3072 | -12.5452 | 2006-3 | 2008-10 |
| AU-Gin | Gingin | 115.7138 | -31.3764 | 2011-11 | 2021-6 |
| AU-GWW | Great Western Woodlands | 120.6542 | -30.1914 | 2013-2 | 2022-6 |
| - | Litchfield | 130.7945 | -13.179 | 2015-7 | 2022-5 |
| - | Longreach | 144.3104 | -23.5233 | 2018-11 | 2022-6 |
| - | Otway | 142.81 | -38.525 | 2007-11 | 2010-12 |
| - | Ridgefield | 116.9668 | -32.5061 | 2016-1 | 2022-6 |
| AU-Rig | Riggs creek | 145.576 | -36.656 | 2011-1 | 2017-7 |
| AU-Rob | Robson Creek | 145.6301 | -17.1175 | 2013-9 | 2022-6 |
| - | Samford | 152.8778 | -27.3881 | 2010-7 | 2017-12 |
| - | Silver Plains | 147.0875 | -42.0906 | 2020-1 | 2021-12 |
| AU-Stp | Sturt Plains | 133.3502 | -17.1507 | 2008-9 | 2022-5 |
| AU-TTE | TiTree East | 133.64 | -22.287 | 2012-8 | 2022-1 |
| AU-Tum | Tumbarumba | 148.1516 | -35.6566 | 2002-12 | 2022-5 |
| AU-Wac | Wallaby Creek | 145.1878 | -37.4259 | 2005-9 | 2013-4 |
| - | Warra | 146.6545 | -43.095 | 2013-4 | 2021-9 |
| AU-Whr | Whroo | 145.0294 | -36.6732 | 2012-1 | 2022-6 |
| AU-Wom | Wombat | 144.094 | -37.4225 | 2010-2 | 2021-5 |
| AU-Yac | Yanco | 146.2908 | -34.9878 | 2013-2 | 2022-6 |

**Figure A1. Locations of 'OzFlux' eddy covariance flux tower sites used in this study. The table on the right liststhe location, start and end dates of the time-series, and the Fluxnet ID for the site where it is available.. The 'stamen' basemap is provided by OpenStreetMap.**

**Table A1: Summary table of the comparison datasets used in the study. Spatial and temporal resolution refers to the extents used by this study, and not necessarily the native ranges. For example, the observation products have been resampled to 0.01 degree, and most datasets have been clipped to 2003 to match the beginning of AusEFlux.**

| Dataset Name | Dataset type | Spatial resolution | Temporal range | References |
|---|---|---|---|---|
| CABLE-POP | Process-model | 1º | 2003-2020 | Friedlingstein et al. (2022) |
| CABLE-BIOS3 | Process-model | 0.25º | 2003-2019 | Villalobos et al. (2022) |
| OCO-2 Inversion | Atmos. inversion | 0.8º | 2015-2019 | Villalobos et al. (2022) |
| GOSAT Inversion | Atmos. inversion | - | 2009-2018 | Metz et al. (2023) |
| FLUXCOM-Met | ML upscaling | 0.5º | 2003-2015 | Jung et al. (2020) |
| FLUXCOM-RS | ML upscaling | 0.083º | 2003-2015 | Jung et al. (2020) |
| MODIS-GPP | Obs. Based | 0.01º | 2003-2021 | Running et al. (2015) |
| GOSIF-GPP | Obs. Based | 0.01º | 2003-2021 | Li and Xiao (2019) |
| DIFFUSE-GPP | Obs. Based | 0.01º | 2003-2021 | Donohue et al. (2014) |

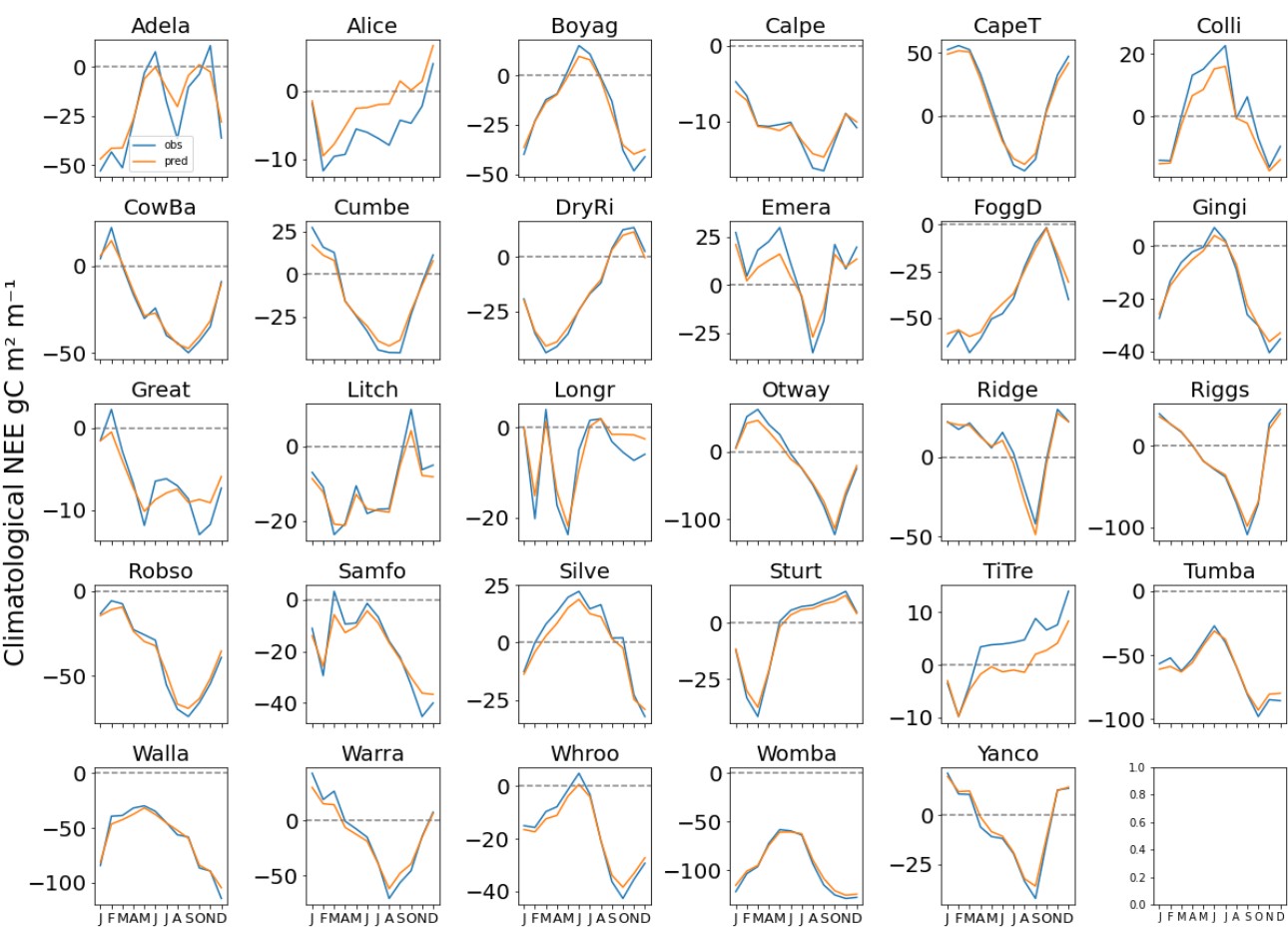

**Figure A2. Climatological seasonal cycles of NEE for each EC flux tower site used in this study, plotted along with the seasonal cycle of the predictions from the nearest pixel to the tower.**

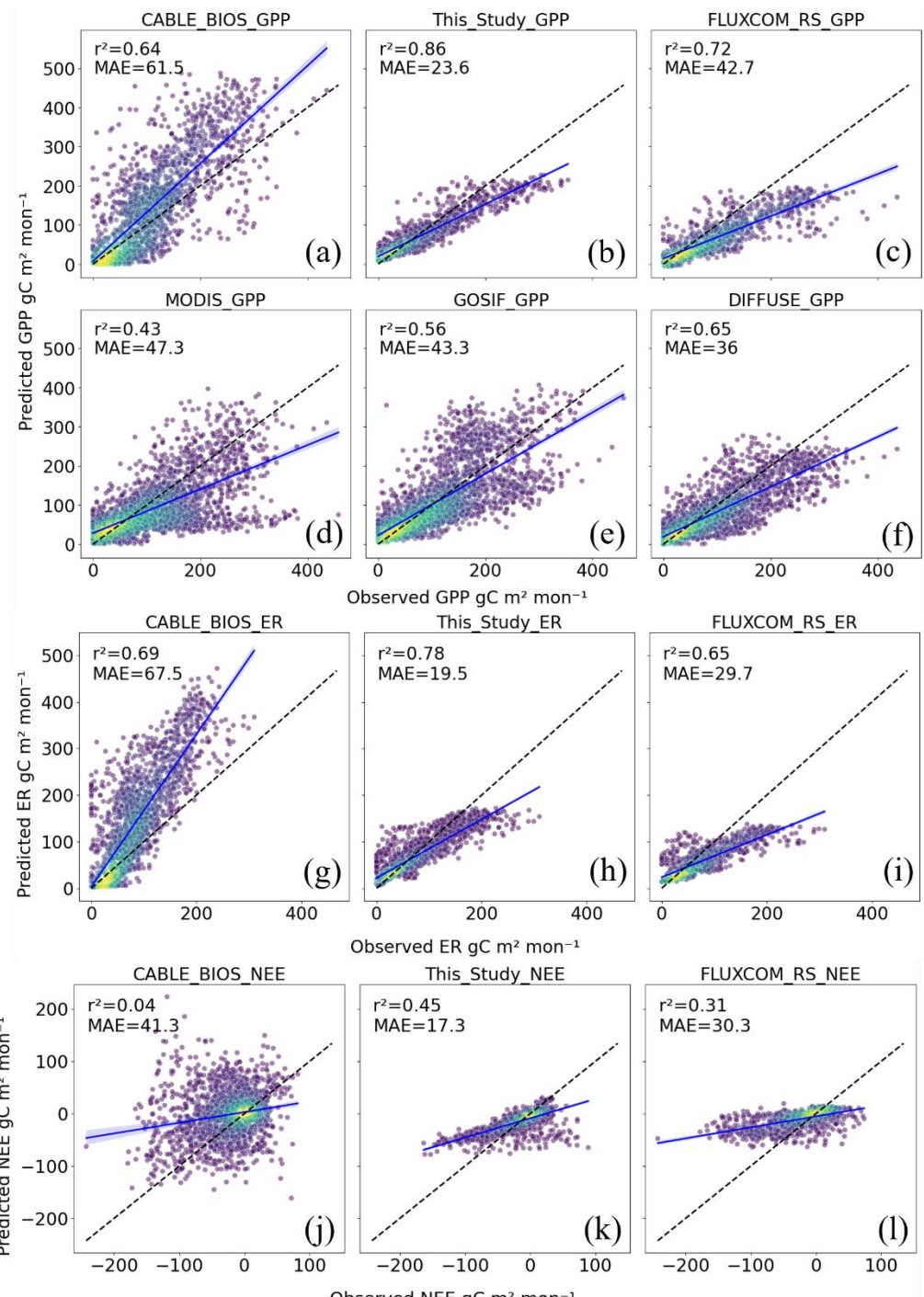

**Figure A3. Scatter plots of modelled vs EC flux tower monthly carbon fluxes for a suite of products. The EC tower flux values are compared with the nearest pixel in each product, and the products have been reprojected to match the resolution of CABLE-BIOS3 (~25 km). Only those products with a reasonably high spatial resolution have been compared with the flux tower (i.e., CABLE-POP, FLUXCOM-Met, and the OCO-2 Inversion have been excluded from these plots).**

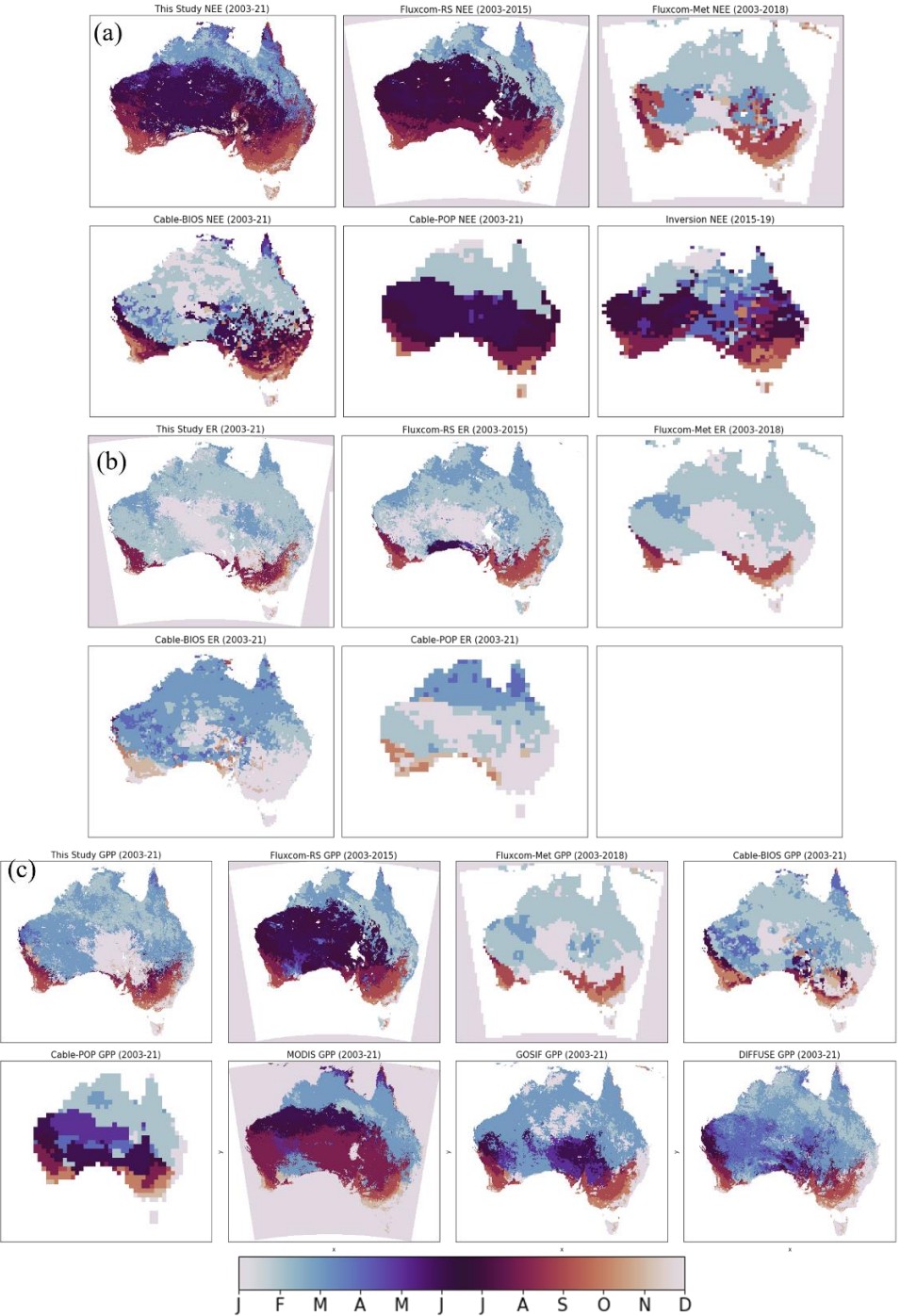

**Figure A4. Climatological month of maximum flux plots. In the case of NEE (b), the pixels show the month of the most negative**
**value (i.e. largest carbon sink). Climatologies are calculated from 2003 and extend to the full length of the available time-series for**
**each product, indicated in the subtitle of each plot.**

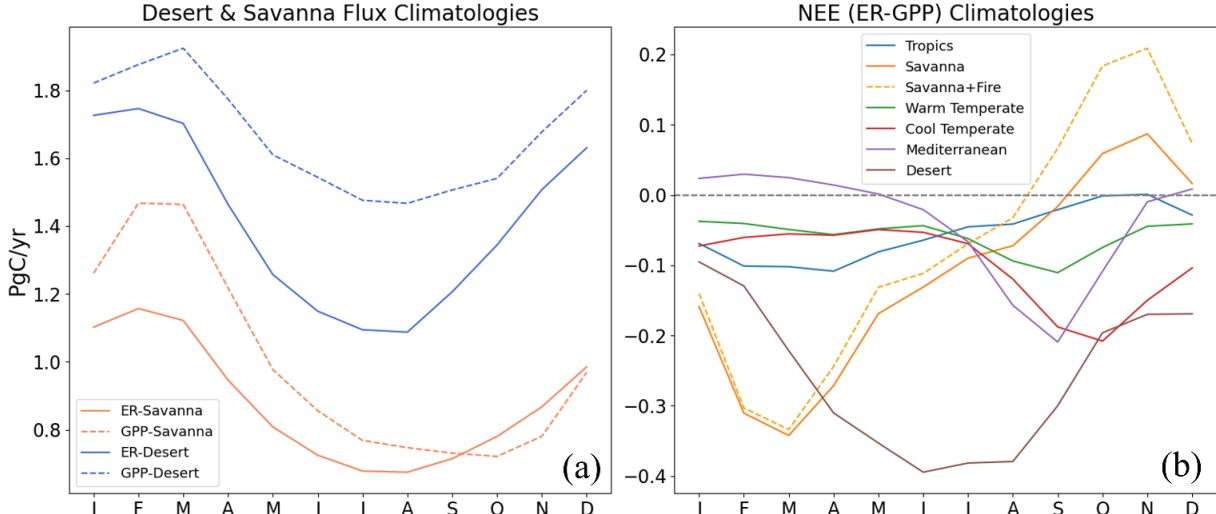

**Figure A5.** (a) Flux climatologies for the Savanna and Desert region, showing the same results as those in Figure 10, but shown on a single plot to enhance interpretability. (b) NEE per bioclimatic region calculated by subtracting GPP from ER (i.e., not directly modelled), presented here to show how the fluxes interact to produce NEE. Fire emissions from the GFAS product have been added to the Savanna fluxes in (b) to highlight how dry season fires interact with ER to create a pulse of carbon to the atmosphere.

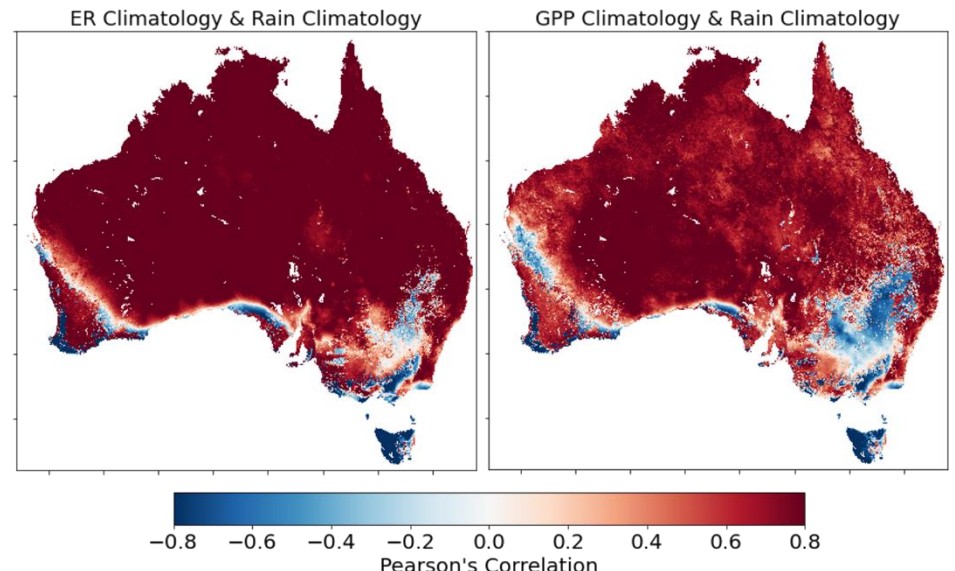

**Figure A6.** Per pixel temporal Pearson correlations between ER climatologies and rainfall climatologies.

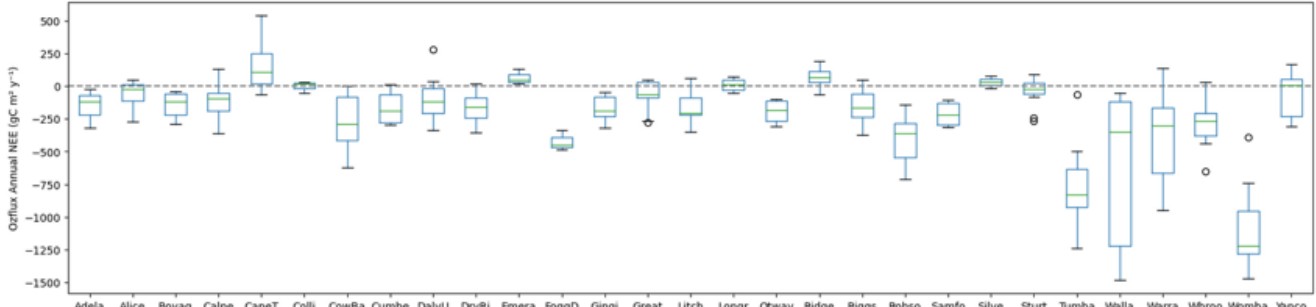

**Figure A7. Boxplots of annual cumulative NEE for each of the sites used in the empirical upscaling.**

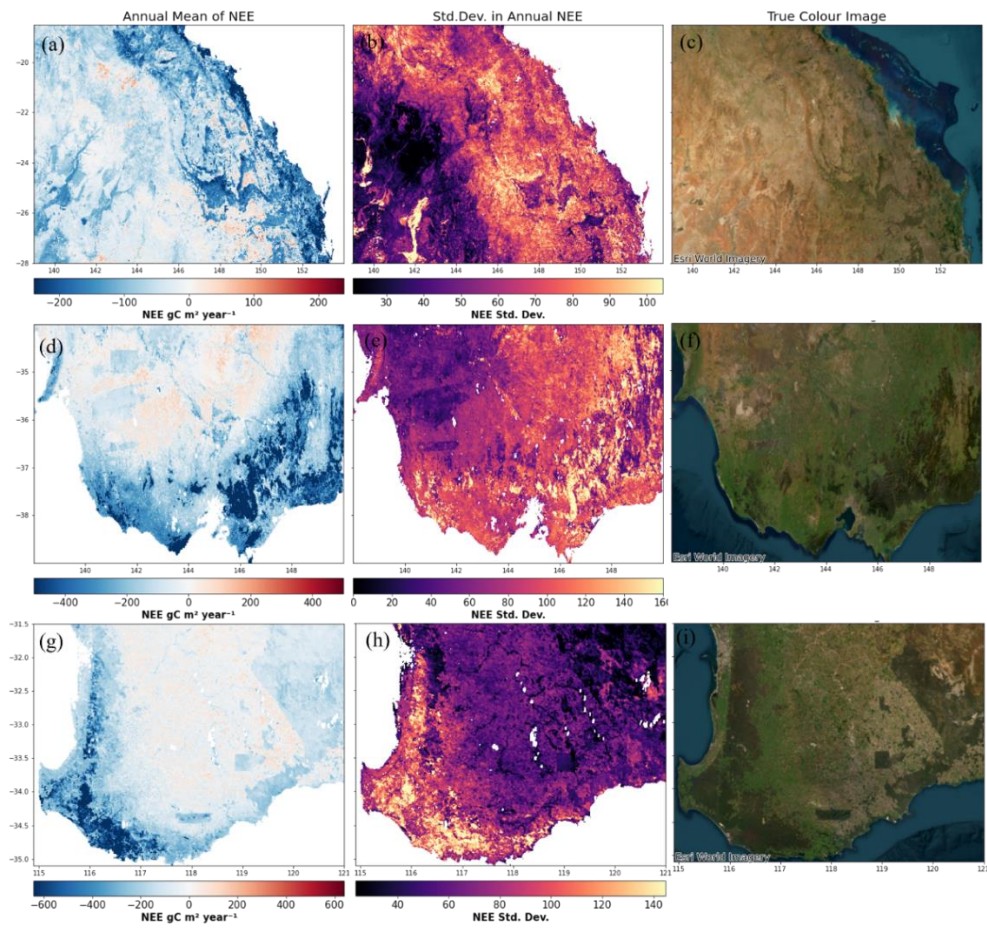


**Figure A8. Maps of annual mean NEE and standard deviation in annual mean NEE zoomed in on three regions to show the landscape features resolved by a high-resolution (1km) dataset of NEE. The top three panels show a region in central Queensland. that extends from the episodic rivers in the south-east (e.g., Coopers Creek), to Townsville in the north west. Panel (c) shows a true colour satellite image (sourced from Esri World Imagery), panel (a) shows the long-term annual mean, and (b) shows the standard deviation in the annual means. Panels d-f show the same south-east Australia extending from Adelaide in the west to Mallacoota in the west. Panels g-f show the same but for south-west Western Australia.**


**Table A2. The hyperparameter grids used during model optimization of the random forest and gradient boosting models. During model fitting, a random grid search was conducted with 250 iterations to identify the highest performing set of hyperparameters.**

| Model | Parameter Grid |
|---|---|
| LGBM | 'num_leaves': stats.randint(5,40), 'min_child_samples':stats.randint(10,30), 'boosting_type': ['gbdt', 'dart'], 'max_depth': stats.randint(5,25), 'n_estimators': [300, 400, 500], |
| RF | 'max_depth': stats.randint(5,35), 'max_features': ['log2', None, "sqrt"], 'n_estimators': [200,300,400,500]} |

**Code Availability.**

The code used to conduct all analysis shown in this manuscript is available on the open-source repository: https://github.com/cbur24/NEE_modelling

**Data Availability.**

The surface gridded carbon fluxes are available from the Zenodo repository at: https://doi.org/10.5281/zenodo.7947265. These fluxes have been resampled to a 5 km grid to facilitate easier uploading and sharing. Full resolution datasets can be provided on request (Burton, 2023).

The Level 6 Ozflux eddy covariance data used by this study is accessible through the Terrestrial Ecosystem Research Network THREDDS data portal, available at: https://dap.tern.org.au/thredds/catalog/ecosystem_process/ozflux/catalog.html. This study relied on the data version "2022_v2", and in instances where both "site-pi" and "default" versions of the datasets were available, we utilised the "default" datasets. See Figure A1 for a full list of sites used.

**Author Contributions.**

CB and LR conceived the study, CB performed all analysis and drafted the manuscript. SR, AVD, and LR provided extensive intellectual input and provided extensive edits to the manuscript.

**Competing interests.**

The authors declare that they have no conflict of interest.

**Acknowledgements**

The authors would like to thank the Terrestrial Ecosystem Research Network (TERN) Ecosystem Processes team, along with the OzFlux site principal investigators whose efforts in collecting and curating the eddy covariance data provides an invaluable

resource to the research community. We would also like to thank the the Terrestrial Ecosystem Research Network (TERN) infrastructure, which is enabled by the Australian Government's National Collaborative Research Infrastructure Strategy (NCRIS). We thank Dr Yohanna Villalobos for providing the CABLE-BIOS3 and the OCO-2 Inversion datasets used in the intercomparison. We recognise the efforts of Dr Randall Donohue who provided access to several datasets and valuable intellectual discussion. Lastly, we thank the National Computing Infrastructure (NCI) which provides a research compute

environment without which this work would not be possible.

**Financial Support**

The first author is supported by a research scholarship provided by Geoscience Australia, funded by the Australian Government.

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
