# Peer review of "Empirical upscaling of OzFlux eddy covariance for high-resolution monitoring of terrestrial carbon uptake in Australia."

_EGUsphere, 2023_

## Author Comment (AC1)

**Review of the Paper: "Empirical upscaling of OzFlux eddy covariance for high-resolution monitoring of terrestrial carbon uptake in Australia"**

**The paper develops high-resolution estimates of GPP, ER, and NEE in Australia using empirical upscaling of flux tower measurements. Comparisons with other products show regional empirical upscaling outperforms global upscaling and process-based models. Rainfall deficits and surpluses drive NEE anomalies, with GPP responding more than ER. The paper introduces "AusEFlux" as a benchmark for high-resolution monitoring of Australia's carbon cycle.**

**Upon careful evaluation and analysis of the manuscript, it is evident that a major revision is necessary in light of the following critical comments. Addressing these concerns will enhance the overall quality and impact of the paper, ensuring its suitability for publication in our esteemed journal.**

- **The paper highlights the performance of regional empirical upscaling in improving global upscaling products and outperforming existing LSMs, but could you provide more insight into the specific limitations of Australia's comparatively sparse network of EC towers and their potential impact on the accuracy of the derived estimates?**

  *We thank the reviewer for the opportunity to provide some more detail on the impact of Australia's sparse EC network on the uncertainty of the AusEFlux estimates. We have edited the paragraph below into the Discussion section of the manuscript.*

  *The principal limitation of the OzFlux EC network is its relatively limited spatial sampling of all the landcover types in Australia. Furthermore, each bioclimatic region is not equally represented, leading to biases in the sampling. For example, desert and xeric ecosystems cover nearly half of the Australian land mass, yet less than 10 % of the network is in these regions (Beringer et al. 2016). Agricultural cropping ecosystems are also under-represented (this is probably why the uncertainty plots of Figure 6 show the greatest variance between predictions is in these regions). Owing to the limitations of the OzFlux network in spatial representation of ecosystems, it is a challenge to confidently claim that cross-validation alone provides an accurate estimate of terrestrial carbon fluxes. As such, we rely heavily on an intercomparison between products as being more indicative (albeit qualitatively) of spatial uncertainty, as we believe the convergence of results from multiple, independent lines of evidence tells us more about the true nature of Australia's terrestrial carbon cycle than any given cross-validation method. We are encouraged by the convergence of our results with the GPP estimates from MODIS, GOSIF, and CABLE-BIOS as each of these datasets applies a different method to quantify GPP. Ecosystem respiration (ER) is harder to effectively validate as only FLUXCOM (similar method to ours) and CABLE provide estimates of ER. Net ecosystem exchange (NEE) offers the prospect of independent validation as*

*the satellite assimilated atmospheric inversions are a wholly independent measurement of NEE (though they still contain significant uncertainties owing to the uncertainties in the satellite $CO_2$ measurements themselves, along with the atmospheric transport model used). This is why we include the two most recent regional-scale inversions in our intercomparisons. Although mean NEE varied between our estimate and those of the two atmospheric inversions, anomalies and the seasonal cycle show better agreement than other methods. We take as evidence that our empirical upscaling of the OzFlux network provides a better estimate of Australia's terrestrial carbon cycle than the global empirical upscaling product, FLUXCOM, which to-date has been the only product available of its type for Australia.*

- **How were datasets resampled to monthly resolution and reprojected to 1\*1 km in section "2.1.2 Gridded explanatory variables"? what was the raw data specifications?**

*Datasets such as MODIS LST, kNDVI, and NDWI were aggregated (simple averaging) to monthly scale using the mean of all available observations within the given month. Spatial reprojection onto a 1 km x1 km geographic grid (EPSG:4326) using either 'bilinear' or 'averaging' resampling methods depending on the native resolution of the product (bilinear for LST, averaging for the others). We have updated the text with this information in section 2.1.2. These datasets were all downloaded from Google Earth Engine, as described in Table 1. The specific MODIS products used were the MCD43A4 v6.1, and MOD11A1 v6.1*

*The climate datasets (from ANUClimate) are provided at 1-km spatial resolution and monthly temporal resolution so no resampling or reprojection was required.*

*The static variables such as landcover fractions and vegetation height were resampled from their native resolutions using the 'average' of all pixels within a 1 km grid cell. The vegetation height product is provided at native 25-m resolution, and the landcover fractions are provided at 250-m resolution. This information has also been added to the manuscript in section 2.1.2.*

- **Provide additional details or references to explain the 'SOLO' data version used for partitioning NEE into GPP and ER. This will aid readers in understanding the specific data processing steps and methods employed.**

*We understand and agree with the reviewer that the SOLO flux partitioning methods may be of interest to some readers. However, we do not believe the details of the methods are within scope of the study because as: (a) the cited work of Isaac et al. (2017) provided the full detail necessary and is available via the references; and (b) adding a description of the methods within this paper would add a lengthy section of text. Nevertheless, we provide a link to the datasets, which are freely available for download through the Terrestrial Ecosystem Research Network (TERN). And we have added to the text within section 2.1.1 a note that a full description of the partitioning method is available from the stated reference. Within the same paragraph we have also*

*provided links to both the OzFlux website, and the TERN website where datasets can be downloaded.*

- **In Section 2.1.3, it is mentioned that the MODIS-GPP and DIFFUSE-GPP products were resampled to a 1 km resolution to match the resolutions of the ML upscaling product. Could you please provide more details regarding the specific method used for resampling these datasets? It would be beneficial to understand the resampling technique employed to ensure compatibility between different resolutions. Additionally, any information regarding potential implications or limitations of the resampling process would be valuable.**

  *Both MODIS- and DIFFUSE-GPP datasets were resampled to 1-km resolution using the average of all pixels within the 1 km cell. MODIS-GPP is provided at a native 500-m, and DIFFUSE-GPP is provided at 250-m resolution. This information is provided in the text at lines 170-171. We do not believe there to be any significant implications of using cell averaging from the moderate resolution (250- and 500-m) to 1-km. Such resampling is common practice in remote sensing.*

- **Elaborate on the resampling and reprojection of gridded explanatory variables. Specify the resampling resolution and provide a rationale for selecting a common 1-km x 1-km geographic grid. Discuss potential errors or limitations associated with spatial resampling and its impact on the accuracy or comparability of the datasets.**

  *We understand the reviewer's point here and refer them to the previous comment above, as some information in response has been given. The rationale for using a 1 km grid is twofold: one, (and the primary reason) is it matches the coarsest native resolution explanatory variables, namely the climate datasets; and two, because the footprint of a flux tower is on the order of 1 x 1 km (very approximately), so it makes sense to extract training data at resolutions comparable to the flux footprint. We have added this short rationale to the 2.1.2 section.*

- **In Section 2.1.3.3, it is mentioned that the regional inverse modeling product by Villalobos et al. (2022) provides a spatial resolution of approximately 81 km. Could you please provide details on how the other datasets with different resolutions were processed and plotted to ensure compatibility for comparison? Specifically, how were the ML results, MODIS-GPP and DIFFUSE-GPP products, which were resampled to 1 km resolution, handled in the analysis?**

  *The ML results are predicted at 1 km spatial resolution, and monthly temporal resolution so were not post-processed (except for some masking of cities). The MODIS and DIFFUSE GPP products were resampled as per the text in section 2.1.3.4. and no further post-processing was done. For the scatter plots of Figure A3, a simple*

*extraction of the pixels over the EC tower locations was done for comparison of the different GPP products against flux tower estimates.*

*For the time-series plots in figures 9, the datasets are all converted to PgC/year from their respective native units (usually gC/m²/month). This involved converting the datasets from their native grid to an equal-area grid of the same spatial resolution (we used EPSG:3577), then multiplying every pixel by its area, along with the conversion factor from grams to petagrams to arrive at PgC/year. This step is documented in the notebook 7_Compare_products.ipynb available on the github repository linked in the assets for this paper. This processing step is common in the literature, and since some effort has gone into creating well documented Jupyter notebooks, we feel there isn't a pressing need to add these steps to the methods section (at the end of section 2.2.2 we have added an explicit reference to the Jupyter Notebooks which describe all processing and analysis steps). After conversion of all datasets to PgC, we summed across all pixels at each time-step to produce 'zonal' time-series. We consider this step to be common enough to not require specific description in the text. The time-series are then smoothed using a three-month rolling mean, which is stated in the caption. For figure 10, seasonal climatologies are created by grouping common months together (i.e. all the Jan., all the Feb. etc.) in a time-series, and then finding the long-term average for each month. The result is then summed across the continent at each monthly time-step to produce a 'zonal' average seasonal cycle. Again, we argue these processing steps are standard practice and do not require a specific description in the text.*

*To assist the reader in understanding if AusEFlux performs better because of the finer spatial resolution, or because it is intrinsically better even when coarsened to the scale of competing products, we have amended Figure A3 in the manuscript. All products in the inter-comparison scatter-plots have now been reprojected to match the resolution of CABLE-BIOS3 (~25km). The figure as it is shown in the manuscript is reproduced below for convenience.*

[Figure]

*Figure A3. Scatter plots of modelled vs EC flux tower monthly carbon fluxes for a suite of products. The EC tower flux values are compared with the nearest pixel in each product, and the products have been  reprojected to match the resolution of CABLE-BIOS3 (~25 km). Only those products with a reasonably high spatial resolution have been compared with the flux tower (i.e.,CABLE-POP, FLUXCOM-Met, and the OCO-2 Inversion have been excluded from these plots).*

- **Specify a specific website or source where readers can access the CO2 flux tower data used in the study. This will facilitate replication and further exploration of the data.**

*We thank the reviewer for noting this and have added to the text at line 109. It now reads: "These data are processed to Level 6 and are freely accessible through the Terrestrial Ecosystem Research Network (https://portal.tern.org.au/)"*

- **Provide more details on the implementation of random forest regression and gradient-boosting decision tree algorithms, including parameter settings, and more importantly elaborate on how predictions from the ensemble of random forest and GBDT models are combined or weighted as an ensemble learning.**

*We appreciate the reviewer's suggestion of providing more information on the ML methods. As such we have provided sufficient detail in the appendix and via the python notebooks should a researcher wish to know more about the hyperparameters used or to reproduce the results. This information has been added to the text in section 2.2.1. We also note that this information is provided in the notebook 3_Generate_ensemble_of_models.ipynb contained on the github page for this study.*

*However, as we fit 30 distinct models per flux, for a total of 90 unique models, it would be impractical to provide hyperparameter settings for each model configuration in the main text. Instead, in the appendix we have provided the parameter grids over which a random grid search was conducted during hyperparameter optimization (Table A1). In section 2.2.1 we provide details on how the model ensembles were combined. A simple per pixel median is conducted across the 30 gridded predictions for a given flux. The interquartile range ($25^{th}$ and $75^{th}$ percentile) are taken as the uncertainty envelope. There are no weightings, each model and gridded prediction is conducted independently and then combined through the calculations of medians/percentiles. We argue this is quite clearly outlined in section 2.2.1. The notebook 5_Combine_ensembles.ipynb provides the documented code on how this was run.*

- **Clarify the rationale and details behind the iterative training procedure with randomly selected EC sites for uncertainty estimation. How can you ensure that all sites were removed in the 30 repeats? Provide details on how the randomness is controlled to achieve this objective.**

*The reason we selected two sites to remove per iteration was because it balanced the need to significantly alter the training dataset per iteration, while not overly degrading the quality of the model by removing too much data. As some of the site's time-series are relatively short, removing only one site could result in removing only 20-30 samples from the training data (from a total of ~2800), and thus the model may only be marginally different from the full-dataset model. Removing more than two sites could result in some iterations where so much data is removed from the training dataset that the quality of the predictions is severely degraded. We have added a statement to section 2.2.1 to clarify this point.*

*It was not our objective to ensure all sites were removed during iteration of the models. Rather, we elected for a uniform random approach where sampling two sites, fifteen times, was merely likely to remove every site. To ensure every possible combination of sites is removed would require $29^2=841$ permutations per flux, which would be impractically time consuming to run, and would be unlikely to tell us much more about the uncertainty than the approach already described. We concede that fifteen iterations are an arbitrary number, but it provides a balance between allowing for a reasonable chance for all sites being removed once, while also keeping the computation time to within reason. The other important consideration is the difficulty in calculating per-pixel percentiles across more than 30 predictions, as each gridded prediction is equivalent to 12.5GiB of data (so already we are summarising close to 400 GiB of data to calculate the ensemble medians).*

- **Provide a detailed description of the data split methodology used in the nested, time-series-split cross-validation approach. Did you consider the aspect of time when splitting and testing the methods? (e.g. did you allocate 5 years for training and 1 or 2 years for testing?)**

*We understand the reviewer's suggestion of providing more information on the temporal cross-validation methods. The time-series split method blocks the testing samples by continuous lengths of time. The exact length of time tested depends on the length of the overall time-series as we allocated 20 % of a timeseries to testing, and 80 % to training. For example, if a dataset is 10 years long, then 8 years is used for training, while a two-year continuous block is used for testing. As we conducted five-fold cross-validation, this procedure was repeated five times and at each iteration the two-year testing 'block' is moved forward in time, such that over the 5-folds, the entire time-series is tested. We have added this example to the text in section 2.2.2. to clarify the method. Also, note that every flux tower record is included in a k-fold, so 20 % of every flux record is tested per fold.*

*The 'nested' part of the CV procedure refers to using a separate, internal split on an outer k-fold to conduct hyperparameter optimization. Using a nested approach to CV prevents testing on the same data used to tune model parameters, and thus prevents creating overly optimistic CV scores. We have included in the text of section 2.2.2 a sentence discussing this, along with a reference that outlines the benefits of using a nested approach.*

*Overall, we have provided a detailed description of the cross-validation procedure, including figure 2 which presents a schematic of the procedure.*

*We would like to also note that the primary focus of this paper is not on providing novel methods for cross-validation (CV), nor on exploring/testing various approaches to ML upscaling. Rather, we have implemented common methodological procedures for empirical upscaling to produce a higher-quality estimate of Australia's terrestrial*

*carbon cycle than already exists (taking advantage of the expansion in the OzFlux tower network and regional feature layers), so that it can be considered alongside other approaches to quantifying the carbon cycle. This is why most of the discussion in the paper is devoted to the intercomparison between products as we feel consiliences between lines of evidence are more important than the results of any given CV method.*

- **Consider incorporating any additional limitations or uncertainties associated with the data sources, processing steps, or comparison datasets. This will provide a more comprehensive understanding of the potential impacts on the study's results and conclusions.**

*We agree with the reviewer that all the products used as explanatory variables in the model are subject to uncertainties. However, the datasets employed by this study are widely used and accepted in the literature. The remote sensing explanatory variables (either MODIS or MODIS-derived) have been widely used at continental scales and their errors have been well documented elsewhere. The interpolated climate data is subject to uncertainties due to the distribution of the measurement network, which in Australia is skewed towards the coast. However, ANUClimate is a well-regarded dataset (Hutchinson et al. 2004, 2014 & 2015) and the density of weather records over Australia is very good during the modern era.*

*We consider describing the uncertainties associated with the inter-comparison datasets as beyond the remit of this paper as it would take considerable time and effort to summarise uncertainties from nine other products. We argue the interested reader can follow-up with the citations provided.*

- **Why was the model not tested on individual sites after training? It is crucial to determine whether the model can perform effectively at a single location.**

*We agree with the reviewer that it is important to assess the model's ability to predict at each flux tower location. However, we believe the manuscript has amply described the model's performance in this respect.*

*Figure A2 compares the predicted seasonal cycles with the observed seasonal cycle for every site in the training dataset (only for NEE as NEE seasonality was a key focus of the paper).*

*In addition to the cross-validation plots of figure 3a-c, we have also included scatter plots of the predicted annual means (Figure 3d-f) and the observed annual means from every site. The colour coding does not show individual sites because distinguishing between 29 unique colours is very difficult, instead we grouped them by bioclimatic regions to make the analysis more legible.*

*We do not see a need to include a figure showing the full time-series predictions of every site as it would make for a very large and unwieldy figure and we feel would not*

*provide the reader with any more useful information than the results shown in Figure 3, and Figure A2.*

**References**

Beringer, J., Hutley, L. B., McHugh, I., Arndt, S. K., Campbell, D., Cleugh, H. A., ... & Wardlaw, T. (2016). An introduction to the Australian and New Zealand flux tower network–OzFlux. *Biogeosciences*, *13*(21), 5895-5916.

Hutchinson, M. F., & Xu, T. (2004). ANUSPLIN version 4.4 user guide. *Centre for Resource and Environmental Studies, The Australian National University, Canberra*, *54*.

Hutchinson, M., & Xu, T. (2014). Methodology for generating Australia-wide surfaces and associated grids for monthly mean daily maximum and minimum temperature, rainfall, pan evaporation and solar radiation for the periods 1990–2009, 2020–2039 and 2060–2079. *NARCliM Report to the NSW Office of Environment and Heritage*.

Hutchinson, M. F., Kesteven, J. L., Xu, T., Evans, B. J., Togashi, H. F., & Stein, J. L. (2015, December). Fine Scale ANUClimate Data for Ecosystem Modeling and Assessment of Plant Functional Types. In *AGU Fall Meeting Abstracts* (Vol. 2015, pp. B43G-0631).

Isaac, P., Cleverly, J., McHugh, I., Van Gorsel, E., Ewenz, C., & Beringer, J. (2017). OzFlux data: network integration from collection to curation. *Biogeosciences*, *14*(12), 2903-2928.

Villalobos, Y., Rayner, P. J., Silver, J. D., Thomas, S., Haverd, V., Knauer, J., ... & Pollard, D. F. (2022). Interannual variability in the Australian carbon cycle over 2015–2019, based on assimilation of Orbiting Carbon Observatory-2 (OCO-2) satellite data. *Atmospheric Chemistry and Physics*, *22*(13), 8897-893

---

## Author Comment (AC2)

I was excited to receive the invitation to review this manuscript, as it is a much needed piece of work that contributes to improved understanding of Australia's terrestrial carbon cycle. It's great to see the use of the extensive OzFlux dataset for validation of the new AusEFlux product, as it is no trivial effort to keep sites running, collate and share eddy covariance data on a regular basis for use in these types of studies. The manuscript itself was very well written and easy to follow, with nicely designed figures and tables (I particularly loved figure 8!) - I applaud the authors on these aspects of their manuscript.

While I enjoyed reading this manuscript and feel it is an important contribution to the research community, I found the discussion section was very limited. First, I was looking for more critique on how the empirical upscaling approach has improved on previous methods, including a well-articulated argument for why regional empirical upscaling needs to be considered by the modelling community to ensure regions are correctly represented in global estimates. There is a heavy bias of flux sites and model parameterization from the temperate northern hemisphere, with regions such as Australia, South America and Africa lacking on-ground validation sites to adequately verify global models. What I really like about this study is that it has elegantly shown there are regional differences in Australia not being adequately captured by global models. If this is true for Australia, surely it could be true for other regions less represented too. I'd like to see more thought and critique on this point in the discussion.

*In the introduction to our manuscript we argue that models built on global datasets and with a strong northern hemisphere bias may not accurately represent ecosystem dynamics in regions where ecosystem responses do not conform to the dominant dynamics in the global dataset. Perhaps owing to the unusual dynamics of the sclerophyllous, evergreen, woody species that dominate Australia's land mass we believe we have proven this hypothesis to be true in the case of Australia. Whether or not this is true for other under-represented regions such as Africa and South America will likely depend on the extent to which their dominant plant species and land cover types conform to the global datasets. Fortunately for Australia, we were able to test this hypothesis for a number of reasons. Firstly, the OzFlux network is reasonably comprehensive in its coverage of Australian ecosystems. Secondly, the EC sites have been 'harmonised' through the implementation of a single dataset standard (Isaac et al 2017) allowing them to be ingested into a single modelling framework. And lastly, the global upscaling product, FLUXCOM, had not included many sites from Australia so differences between FLUXCOM and AusEFlux are likely attributable to differences in the training data. For Africa and South America a set of high quality, harmonised EC sites covering the diversity of their ecosystems does not appear available, thus it may not be possible to test if these regions are being misrepresented by global models.*

*To highlight these points we have added a short paragraph to the discussion section that read as follows:*

*"Our study showed that increasing the diversity of flux tower sites beyond the small Australian set used in global products improved the quality of carbon flux estimates. We cannot predict whether the same might hold for other underrepresented regions, which mostly coincide with the global south, or whether the isolated evolution of Australia's ecosystems also plays a role."*

**There's also a lack of discussion of how the limitations in this study could be overcome. For example, the method itself seems sound, but expanding on the points about more EC data being needed would be really helpful to the research community. Do the authors feel that just longer timeseries from the current network are needed, or are more sites required? If more sites, where are they needed? Figure 8 showed some interesting GPP and ER dynamics in certain areas where there is a distinct lack of EC observation sites, the WA Wheatbelt being one of them. The authors point to these areas in the results section, but do not address how these areas could be better understood by future research efforts. A better discussion around these points, in a nationally focused paper like this, would really help researchers on the ground level to make the case for the need to fill these missing gaps.**

*We agree with the reviewer that some discussion on where future OzFlux sites should be placed is worth adding into the manuscript, and to this end we have edited this discussion into a broader paragraph discussing some of the uncertainties with spatial sampling of OzFlux (the entire paragraph is quoted below).*

*"A limitation of the OzFlux network is the necessarily limited repeat spatial sampling of all main land cover types. Furthermore, not each bioclimatic region is equally well represented, leading to biases in the sampling. For example, desert and xeric ecosystems cover nearly half of the Australian land mass, but less than 10% of the sites are located in these regions (Beringer et al., 2016).  Australia's expansive cropping ecosystems are also under-represented. The limited representation of these systems in the training data is likely why we found comparatively high uncertainty in these regions (Fig. 6). Further uncertainty in the cropping regions may also be due to the heterogeneity of crop types and agricultural practices that may not be represented in our feature layers, and the potentially large carbon exports as agricultural commodities.  Also, given the Australian Government's emphasis on emission offsetting through changes in agricultural practices and human-induced regeneration of native woody vegetation, especially in drier regions (DCEEW, 2023), new EC sites in cropping regions and in the (semi) arid rangelands areas of New South Wales, Queensland, and Western Australia might help reduce uncertainties in AusEFlux and expand the evidential basis for carbon sequestration through (re-)vegetation (Macintosh et al., 2022). Given the changing climate conditions of Australia, it is vital to maintain the current OzFlux infrastructure so that future changes to climate-carbon interactions are monitored at the continental level through iterative retraining of the AusEFlux model as new data is collected."*

**Lastly, there's a lack of discussion around future directions for this work. There's momentum building and wider interest in understanding carbon fluxes from landscapes in real time and in collating annual budgets at national scale more frequently. There's an opportunity for the work presented in this manuscript to be incorporated into a regularly produced national annual estimate of Australia's terrestrial carbon accounting, but there's no mention of this in the discussion. I suggest the authors consider adding text along these lines, perhaps pointing to the vision outlined in Papale 2020 and efforts already underway to deliver national carbon observing infrastructure, such as TERN, NEON (USA) and ICOS (EU), that could be input into approaches like the one presented by the authors to help realise these goals.**

*We agree with the reviewer on this point and thank them for prompting further discussion on the future directions of the work.  Please see the comment below in dot point  "***Lines 467-469: …***" for our response to the argument on this work being incorporated into a regularly produced national estimate of Australia's terrestrial carbon budget.*

*We have also added further information on future directions for this work at the end of the discussion section. The last paragraph of the discussion now reads:*

*"While our estimate provides a step-forward in our means for assessing the complex, seasonal, and interannual dynamics of Australia's carbon cycle, future work can improve upon this current effort.  Firstly, we aim to extend AusEFlux further back in time through the inclusion of satellite observations from the AVHRR and Landsat missions. However, this effort will inform a separate study as it will require solving cross-sensor calibration issues.  A longer record of empirically derived terrestrial carbon fluxes will assist in defining robust environmental baselines from which future changes to the carbon cycle can be assessed. Secondly, new or improved feature layers can be incorporated as they become available (e.g., time-varying estimates of the percentages of trees, grass and bare). And lastly, we aim to explore the prospects of ecological forecasting (Dietze et al., 2018) of the terrestrial carbon cycle as seasonal forecasts may be possible where forecasts of the climate are sufficiently detailed."*

**There are a few other specific items I feel need to be addressed before the manuscript is ready for publication. I've identified these as follows and believe that if the authors can address them, their manuscript will be more widely cited as a result:**

**- Lines 37-45: This is quite a difference between the two studies, but then at line 55 it's revealed that the Villalobos et al. 2022 study was from years 2015-2019, while the Friedlingstein et al. 2022 study was from 2003-2021. Looking up both studies reveals these time frames to be accurate. While I completely agree that regionally forced studies usually provide more accurate estimates of carbon cycling in Australia, I think it is misleading not to mention the temporal mismatch between these studies. I suspect the temporal mismatch could be the primary cause of the difference of >50 % between studies, as the millennium drought**

**(2001-2009) would be captured in the Friedlingstein et al. anaylsis but not in the Villalobos et al analysis. Please amend the text to take this into consideration.**

*The references for this section has led to some confusion, for which we apologise. Here we are comparing CABLE-POP (extracted from TRENDY v10) with CABLE-BIOS3, provided by Villalobos (2022). Both datasets were clipped to the 2003-2019 range (we incorrectly stated the range as 2003-2021 in the manuscript and have corrected this) for the comparison between long-term mean GPP, thus there is no temporal mismatch. In the Villalobos reference their atmospheric inversion only ran from 2015-2019, but their run of CABLE extends much longer. We have amended the text slightly to make this more clear. We have also included a table in the appendix that outlines the main features of each comparison dataset, which should also help reduce confusion.*

**- Line 57; Please add spatial resolution for better comparison with OCO-2, i.e. as at lines 54-55.**

*The Metz (2023) dataset is averaged over the entire Australian TRANSCOM region (including NZ) and is provided as a zonally summarised time-series so there is no spatial resolution given. The initial $CO_2$ fluxes were resampled to 1 x1 degree grid before applying their TRANSCOM region mask, but it is not clear to us from their supplementary material if the atmospheric inversion results were first predicted on a grid and then summarised, or summarised and then predicted.*

**- Line 70: I think its important to identify here the unequal representation of EC sites across the globe, as some biomes (i.e. the tropics) contain a limited number of sites compared to the temperate northern hemisphere. This bias is also likely to be affecting ML empirical upscaling approaches. I see the authors allude to this at line 78, but I think it needs to be addressed here too. See Baldocchi et al. 2018 (https://doi.org/10.1016/j.agrformet.2017.05.015) for a good review of inter-annual variability in NEE from sites around the world, and where long-term monitoring sites are lacking.**

*We agree that the northern hemisphere bias in the Fluxnet dataset is an important attribute to highlight, and that mentioning this earlier in the manuscript during the discussion on the limitations of FLUXCOM is worthwhile. We have included the following sentence in that section:*

*"The global FLUXNET2015 dataset is also biased to the northern hemisphere, which may preclude global upscaling products from making quality predictions in regions that are both underrepresented in the training data, and do not conform to northern hemisphere climate dynamics."*

**- Line 77: A better introduction citation for the OzFlux network is Beringer et al. 2022 ( https://doi.org/10.1111/gcb.16141) or Beringer et al. 2016 (https://doi.org/10.5194/bg-13-5895-2016). Isaac et al. 2017 is an excellent publication to cite for how the flux data were processed, which should be in the methods.**

*We have included the Beringer 2016 & 2022 references in this location.*

**- Line 108: Please add the following text here to clarify how the data were processed "using PyFluxPro vXXX (Isaac et al. 2017),..." The authors may need to check with TERN regarding the PyFluxPro version used.**

*We argue that providing the reader with the specific PyFluxPro versions used to process the Level-6 data will not be valuable to the reader unless they are already aware of what the software iterations mean. However, we accept the more general point underlying this comment about reproducibility of AusEFlux, and to this end we have included specific reference to the versions of the datasets used. We have also included URL paths to find the data, and Table A1 has been updated to include start and end dates of the datasets used. The beginning of section 2.1.1 now reads:*

*"We used monthly fluxes of NEE, GPP, and ER produced by the OzFlux (https://ozflux.org.au/) regional network of eddy covariance flux towers. These data are processed to Level 6 and are freely accessible through the Terrestrial Ecosystem Research Network THREDDS portal (https://dap.tern.org.au/thredds/catalog/ecosystem_process/ozflux/catalog.html (TERN, 2023). All site data used in this study were version "2022_v2", and in instances where both "site-pi" and "default" versions of the datasets were available, we utilised the "default" datasets."*

**- Lines 189-191: Can the authors comment on this more specifically? Are there any biomes or land uses missing that in their opinion would make the analysis more robust? Perhaps this could come in the discussion instead...?**

*We have included in the discussion section more comments on the limitations of the training data, including where we believe further EC sites would help reduce uncertainties that are derived from the training data.*

**- Lines 321-325: I agree with this statement, but it should appear in the discussion, not results. Please move to discussion, a good place would be the final discussion paragraph.**

*We agree and have moved these statements to a new section at the beginning of the discussion section.*

**- Lines 406-422: This paragraph is mixing results and discussion a bit, i.e. lines 408-410 and lines 415-416, Please consider moving these points to the discussion, which would help beef up the section.**

*While in general we agree with the reviewer that mixing results and discussion is an issue, in this case we argue that there is merit in including these brief explanations for the results as it assists the reader in understanding discrepancies between products at the point of encounter. As the 'discussion' parts of this section only amount to two*

*sentences, we argue there is utility in keeping these comments where they currently are in the manuscript because it improves clarity.*

**- Line 438: Remind readers of this study here, it's the Villalobos et al. 2022 study, correct? In fact, it would be useful to include a small table that includes information about each of the models used in this study, who published them, their general characteristics (temporal and spatial resolution), etc... That way the authors can refer the reader here to table X for a refresh and avoid re-citing each study, that would add clutter to the text below. The table could be a brief summary of information presented in section 2.1 and included at the end of that section.**

*We appreciate the suggestion of adding a table to summarise the comparison datasets as there are quite a few products and it does get convoluted at times. We have created a table that includes the dataset name, dataset type (process model, inversion etc.), the spatio-temporal extents, and the key reference for each study. We have added this table (Table A1) to the appendix to avoid cluttering the manuscript with too many tables and figures. We've included a copy of the table below.*

| Dataset Name | Dataset type | Spatial resolution | Temporal range | References |
|---|---|---|---|---|
| CABLE-POP | Process-model | $1^0$ | 2003-2020 | Friedlingstein et al. (2022) |
| CABLE-BIOS3 | Process-model | $0.25^0$ | 2003-2019 | Villalobos et al. (2022) |
| OCO-2 Inversion | Atmos. inversion | $0.8^0$ | 2015-2019 | Villalobos et al. (2022) |
| GOSAT Inversion | Atmos. inversion | - | 2009-2018 | Metz et al. (2023) |
| FLUXCOM-Met | ML upscaling | $0.5^0$ | 2003-2015 | Jung et al. (2020) |
| FLUXCOM-RS | ML upscaling | $0.083^0$ | 2003-2015 | Jung et al. (2020) |
| MODIS-GPP | Obs. Based | $0.01^0$ | 2003-2021 | Running et al. (2015) |
| GOSIF-GPP | Obs. Based | $0.01^0$ | 2003-2021 | Li and Xiao (2019) |
| DIFFUSE-GPP | Obs. Based | $0.01^0$ | 2003-2021 | Donohue et al. (2014) |

**- Lines 438-442 - this is all one sentence, which is long and rather confusing to follow. Please revise and more clearly articulate to the reader that this study was verified using OzFlux EC sites.**

*We agree that this sentence was unwieldy and have rephrased the paragraph.*

*"We found evidence that Australia is, on average, a stronger annual carbon sink than previous CABLE LSM and FLUXCOM estimates have concluded. Our estimate of the long-term annual mean carbon sink over Australia (-0.44 PgC/yr) is higher than those reported by any study besides the regional OCO-2 inversion (-0.47 PgC/yr). We take the consilience between our estimate and the OCO-2 inversion's; the fact that 25 out of the 29 OzFlux EC sites used here report strong annual mean carbon sinks (Figure A7), and the theoretical argument that ML predictions tend to produce good estimates of the mean as evidence that Australia's status as a comparatively strong net carbon sink is robust."*

**- Line 460: Table shouldn't appear in the discussion.**

*Agreed, it's been moved back to the results section below section 3.5.*

**- Lines 467-469: Can the authors expand on this point more? In an ideal world, how frequently do the authors think a product like this should be updated? Realistically, how frequently is this likely to be? I recommend reading Papale 2020 (https://doi.org/10.5194/bg-17-5587-2020) and publications from the global carbon project to tease this discussion point out further.**

*We are happy to expand on this point further, and it is a future aim of the authors to annually update and release this product. We have included in the discussion section the following paragraph:*

*"An advantage of this approach over other methods is its computational efficiency, and, owing to the mature architecture of the OzFlux infrastructure, the ability to programmatically ingest updated or new EC datasets to further refine models. Thus, there is an opportunity for AusEFlux to be incorporated into an annually produced national estimate of Australia's terrestrial carbon fluxes. Any annually produced 'bottom-up' estimate of Australia's terrestrial carbon fluxes could also serve as a compliment to the Global Carbon Project's aims of annually reporting the carbon balance of the world (Papale, 2020). The primary challenge with any operational reporting framework is ensuring the necessary feature layers are updated with a similar cadence, so future work will involve identifying reliable and regularly updated data streams to serve this end. Through regular updating of this dataset, the ecosystems that play an outsized role in controlling Australia's mean carbon sink and contribute substantially to its IAV can begin to be systematically monitored for change."*

**- Lines 477-478: What about the role of fire in consuming biomass in the dry season and how that might affect carbon emissions from savannas? Can the authors expand on this please. Beringer et al. 2015 ( https://doi.org/10.1111/gcb.12686) might be a good place to start.**

*We thank the reviewer for prompting us to discuss the seasonal role of fire. To elucidate the seasonal role of fire in the savanna regions, we have amended Figure A5 to include fire emissions (copied below). The addition of fire emissions shows that the late dry-season (Aug-Oct) fires lead to an earlier net carbon pulse to the atmosphere and larger peak emissions than the out-of-phase ER-GPP effect alone. We have amended the discussion in the manuscript to reflect this.*

[Figure]

*Figure A5. (a) Flux climatologies for the Savanna and Desert region, showing the same results as those in Figure 10, but shown on a single plot to enhance interpretability. (b) NEE per bioclimatic region calculated by subtracting GPP from ER (i.e., not directly modelled), presented here to show how the fluxes interact to produce NEE. Fire emissions from the GFAS product have been added to the Savanna fluxes in (b) to highlight how dry season fires interact with ER to enhance a seasonal pulse of carbon to the atmosphere.*

**- Lines 478-480: Here again, a missed opportunity to critique with site-based studies, such as Cleverly et al. 2013 (10.1002/jgrg.20101)**

*We have added reference to the Cleverly et al. (2013) paper in this section, which now reads:*

*"This finding agrees with Renchon et al. (2018) at the Cumberland Plains EC flux tower site, where the forest was a CO2 sink in winter and a source in summer due to larger seasonal amplitudes in ER. Similarly, Metz et al. (2023) found that seasonal rainfall in semi-arid regions after the dry season drives pulses of heterotrophic respiration that precede the GPP response, leading to net carbon uptake not beginning until March. Cleverly et al. (2013), in a site-based study of a semi-arid acacia woodland in central Australia, observed that the first large springtime storms following the dry season*

*resulted in pulses of ecosystem respiration owing to an uptick in moisture limited microbial decomposition of photodegraded litter and flushing of CO2 from soil pore spaces through infiltration.  Our results confirm that ER over the savanna region responds quickly to seasonal rainfall events at the end of the dry-season, while GPP responds more slowly resulting in carbon pulses to the atmosphere during the Oct-Dec period."*

**- Lines 489-492: This is a rather subjective and negative way to begin a conclusion. One could argue that the OzFlux network already captures a diverse range of Australian ecosystems, and it is certainly the largest network in the underrepresented southern hemisphere. However, one could also argue that there are key systems missing, which can bias any upscaling approaches that use OzFlux data. How many flux towers are needed for a network like OzFlux to have "good" coverage? My point being, given this paper did not assess whether the quantity of sites in OzFlux was adequate for upscaling (in fact it used correlation with OzFlux sites as an indicator that the results were robust), I suggest rethinking the opening sentence of the conclusion to be more focused on the key result/finding and less about the limitations of OzFlux.**

*We thank the reviewer for this suggestion and agree that the opening sentence was needlessly negative. We have deleted the opening line of the conclusion and the conclusion now has a better focus on the key findings.*

---

## Author Comment (AC3)

We'd like to thank the authors for this interesting and novel use of the TERN Ecosystem Processes /OzFlux eddy covariance data. It is especially rewarding to see Australian authors make use of the data, which has been freely shared by site PI's over the last decade or so.

We have a few minor suggestions that we feel would help strengthen the paper, by making the eddy covariance data more transparently accessible to other researchers, clarifying the processing steps used in the flux data, and providing proper acknowledgement of the data sources.

- **Please provide more details of the flux data in section 2.1.1. Readers need to know which server the data was downloaded from. We are guessing this is likely to be the OzFlux THREDDS server, in which case the URL needs to be provided.**

  *We acknowledge that we were a little vague in that section on precisely which versions of the data we used and where we extracted it from, so thank you for pointing out the need for further details . We extracted the data from the TERN THREDDS data portal. We have added this url path to section 2.1.1. We have also included reference to the version of data used in this study: i.e., "2022_v2-default".*

- **Also in Section 2.1.1: Although available as an option (Isaac et al., 2017), MODIS EVI data were not used in the processing pipeline of the default data products the authors appear to have used. The default drivers for the SOLO neural network were air temperature, soil temperature and soil moisture. Please clarify and correct that statement.**

  *Thank you for pointing out this mistake, we have corrected the statement in the manuscript. It now reads: "This study uses the 'SOLO' data version which is calculated using a data-driven nocturnal respiration approach for partitioning where respiration is modelled using an artificial neural network driven by air and soil temperature, and soil water content."*

- **Provide more details of which flux data were used in the Data Availability section at the end of the paper.**

  *We have added the following statement to the Data Availability section:*

  *The Level 6 Ozflux eddy covariance data used by this study is accessible through the Terrestrial Ecosystem Research Network THREDDS data portal, available at: https://dap.tern.org.au/thredds/catalog/ecosystem_process/ozflux/catalog.html. This study relied on the data version "2022_v2", and in instances where both "site-pi" and "default" versions of the datasets were available, we utilised the "default" datasets. See Figure A1 for a full list of sites used.*

- **Add details to the table embedded in Figure A1 to give the start and end dates of the data streams used. Also, please include the official FLUXNET 2015 IDs for the site names, so that the global flux community can understand which sites were used.**

*We have updated the table in Figure A1 to now include the Fluxnet ID (where available), and have also included the full name of the site, and the starting and ending dates of the data used in the study. For space reasons we have removed some of the summary climate information that was previously in the table.*

- **Collective acknowledgement of TERN Ecosystem Processes/OzFlux site PIs in the Acknowledgements section would be greatly appreciated.**

  *We fully appreciate that this work is only possible through the collective efforts of the OzFlux and TERN teams, and we have amended the acknowledgements section to more clearly state this.*

  *"The authors would like to thank the Terrestrial Ecosystem Research Network (TERN) Ecosystem Processes team, along with the OzFlux site principal investigators whose collective efforts in acquiring and curating the eddy covariance data provides an invaluable resource to the research community."*

- **As a courtesy, please consider sending an email to individual PIs, whose contact information is contained in all the netCDF files, advising the use of the data. Alternatively or in addition, feel free to contact the TERN Ecosystem Processes (Lucas Cernusak) lead or the OzFlux director (Jamie Cleverly) as a single point of contact to advise that you are using the data.**

  *We have contacted Lucas Cernusak and Jamie Cleverly and they have responded favourably.*

**We also recognize that there are steps we can take as TERN Ecosystem Processes/OzFlux data providers to make the data more readily accessible and citable. We thank the authors for bringing this to our attention through the useful contribution of their paper.**